# The FAM53C/DYRK1A axis regulates the G1/S transition of the cell cycle

Taylar Hammond[1,2], Jong Bin Choi[1], Miles W Membreño[3], Janos Demeter[4,5], Roy Ng[4,5], Debadrita Bhattacharya[1], Thuyen N Nguyen[1], Griffin G Hartmann[1,2], Caterina I Colon[1,2], Carine Bossard[6], Jan M Skotheim[2], Peter K Jackson[4,5], Anca M Pasca[1], Seth M Rubin[3], Julien Sage[1,7]*

[1]Departments of Pediatrics, Stanford University, Stanford, United States; [2]Department of Biology, Stanford University, Stanford, United States; [3]Department of Chemistry and Biochemistry, University of California, Santa Cruz, Santa Cruz, United States; [4]Department of Microbiology and Immunology, Stanford University, Stanford, United States; [5]Department of Pathology, Stanford University, Stanford, United States; [6]Biosplice Therapeutics Inc, TenaRx Inc, San Diego, United States; [7]Department of Genetics, Stanford University, Stanford, United States

## eLife Assessment

This study identifies the uncharacterised protein FAM53C as a novel, potential regulator of the G1/S cell cycle transition, linking its function to the DYRK1A kinase and the RB/p53 pathways. The work is **valuable** and of interest to the cell cycle field, leveraging a strong computational screen to identify a new candidate. The findings are **solid**, although confidence in the siRNA depletion phenotypes would have been higher with rescue experiments using an siRNA-resistant cDNA.
[Editors' note: this paper was reviewed by Review Commons.]

*For correspondence: julsage@stanford.edu

**Abstract** A growing number of therapies are being developed to target the cell cycle machinery for the treatment of cancer and other human diseases. Consequently, a greater understanding of the factors regulating cell cycle progression becomes essential to help enhance the response to these new therapies. Here, using data from the Cancer Dependency Map, we identified FAM53C as a new regulator of cell cycle progression. We found that FAM53C is critical for this cell cycle transition and that it acts upstream of the Cyclin D-CDK4/6-RB axis and of p53 in the regulation of the G1/S transition. By mass spectrometry, biochemical, and cellular assays, we identified and validated DYRK1A as a cell cycle kinase that is inhibited by and directly interacts with FAM53C. Consistent with the role for FAM53C identified in cells in culture, *FAM53C* knockout human cortical organoids display increased cell cycle arrest and growth defects. *Fam53C* knockout mice show minor behavioral phenotypes. Because DYRK1A dysregulation contributes to developmental disorders such as Down syndrome as well as tumorigenesis, future strategies aiming at regulating FAM53C activity may benefit a broad range of patients.

## Introduction

The cell cycle is precisely regulated by a complex network of checkpoints. Among those, the G1 checkpoint is an essential decision point for cell cycle progression into S-phase or cell cycle exit in G0, either for a transient arrest, such as quiescence, or for a more permanent arrest, such as during cellular differentiation or senescence. Signal integration at this cell cycle checkpoint is tightly regulated, and defects at this transition can have catastrophic impacts in development and human diseases, including

autoimmune diseases and cancer (*Matthews et al., 2022*). A better understanding of the mechanisms regulating the G1/S decision point is critical for our understanding of embryonic development, tissue repair, regeneration, and aging processes.

The G1/S transition is regulated by multiple intracellular and extracellular signals, many of which converge onto the RB pathway (*Sherr, 1994*). The core RB pathway is centered around the retinoblastoma (RB) tumor suppressor. RB normally inhibits cell cycle progression in G1 by interacting with various proteins, including the E2F family of transcription factors, which regulate the expression of genes essential for DNA replication and other aspects of cell division (*Kent and Leone, 2019*). Under the prevailing model, mitogen signaling stimulates RB hyperphosphorylation and inactivation by Cyclin/CDKs (namely Cyclin D-CDK4/6 and Cyclin E-CDK2), leading to E2F release to promote proliferation; mechanisms such as upregulation of the p21 cell cycle inhibitor (p21WAF1/CIP1, encoded by the *CDKN1A* gene) under stress conditions, often downstream of p53 activation (*Engeland, 2022*), can limit CDK activity to keep cells in G1 arrest (*Engeland, 2022*).

Cyclin D (with three family members Cyclin D1, D2, and D3) is a critical Cyclin whose levels are tightly regulated both at the transcriptional level and the post-transcriptional level, including by regulation of protein stability to induce rapid degradation (*Sherr and Sicinski, 2018*; *Saleban et al., 2023*). Recently, we and others have shown that AMBRA1 (Activating Molecule in Beclin1-Regulated Autophagy) is an adaptor for the CRL4 (CUL4-RING) E3 ubiquitin ligase to mediate the polyubiquitylation and subsequent degradation of cyclin D proteins in cells (*Chaikovsky et al., 2021*; *Maiani et al., 2021*; *Simoneschi et al., 2021*). These studies resolved a long-standing question in the field related to the molecular mechanisms regulating Cyclin protein levels regulation, but also highlighted that, despite decades of research, key regulators of core cell cycle machinery remain to be discovered and characterized.

Accumulating evidence also indicates that Dual-specificity Tyrosine Phosphorylation-regulated Kinase 1 A (DYRK1A), and its family member DYRK1B, can phosphorylate Cyclin D (at threonine 286, T286, for Cyclin D1), which leads to lower Cyclin D protein levels in cells (*Chen et al., 2013a*; *Ashford et al., 2014*; *Soppa et al., 2014*; *Thompson et al., 2015*; *Hille et al., 2016*; *Bélanger et al., 2023*; *Lindberg et al., 2023*) while also modulating p21 levels in a bistable manner (*Chen et al., 2013a*). DYRK1A has gained clinical interest due to its dosage-sensitive role in cancer and developmental disease: the *DYRK1A* gene maps to the Down syndrome critical region on chromosome 21q22 (*Rammohan et al., 2022*; *Laham et al., 2021*) and increased DYRK1A kinase activity has been linked to many of the developmental brain defects associated with Down syndrome (*de Souza et al., 2023*; *Wegiel et al., 2011*; *Ryoo et al., 2007*). In addition, functional loss of one *DYRK1A* allele leads to the so-called DYRK1A haploinsufficiency syndrome (*Ji et al., 2015*). *DYRK1A* variants have also recently been associated with epilepsy (*Boßelmann et al., 2024*). The regulation of DYRK1A and its effects toward Cyclin D1 and p21 in normal and disease contexts remain poorly understood.

Here we aimed to identify new regulators of the G1/S transition of the cell cycle. Using co-dependency data in hundreds of human cancer cell lines from the Cancer Dependency Map Project (DepMap, *McFarland et al., 2018*), we identified the poorly studied protein FAM53C as a candidate cell cycle regulator connected to the RB pathway. Our work shows that FAM53C normally promotes G1/S progression and links its cell cycle activity to DYRK1A activity in cells.

## Results

### A DepMap analysis identifies FAM53C as a candidate regulator of G1/S

To identify new candidate regulators of the G1/S switch, we took advantage of the co-dependency scores offered by the Cancer Dependency Map (DepMap) platform to identify potential novel pathway interactors (*Takemon et al., 2023*; *Doherty et al., 2022*). To find novel interactions, we first generated a list of 38 factors playing a known role at this cell cycle transition (*Supplementary file 1*).

We next overlapped the top 100 co-dependencies from the Cancer Dependency Map (DepMap) platform (*Tsherniak et al., 2017*) for each of the input genes to generate a list of shared co-dependency genes from across the input list (*Figure 1A* and *Supplementary file 2*). We established a score for each gene hit based on the number of input genes which shared a co-dependency with the hit. The gene coding for CDK2 had the maximum of 13 hits in the analysis, while genes coding for other established G1/S factors such as E2F1 or Cyclin E2 had a score of 3, establishing a cutoff for new

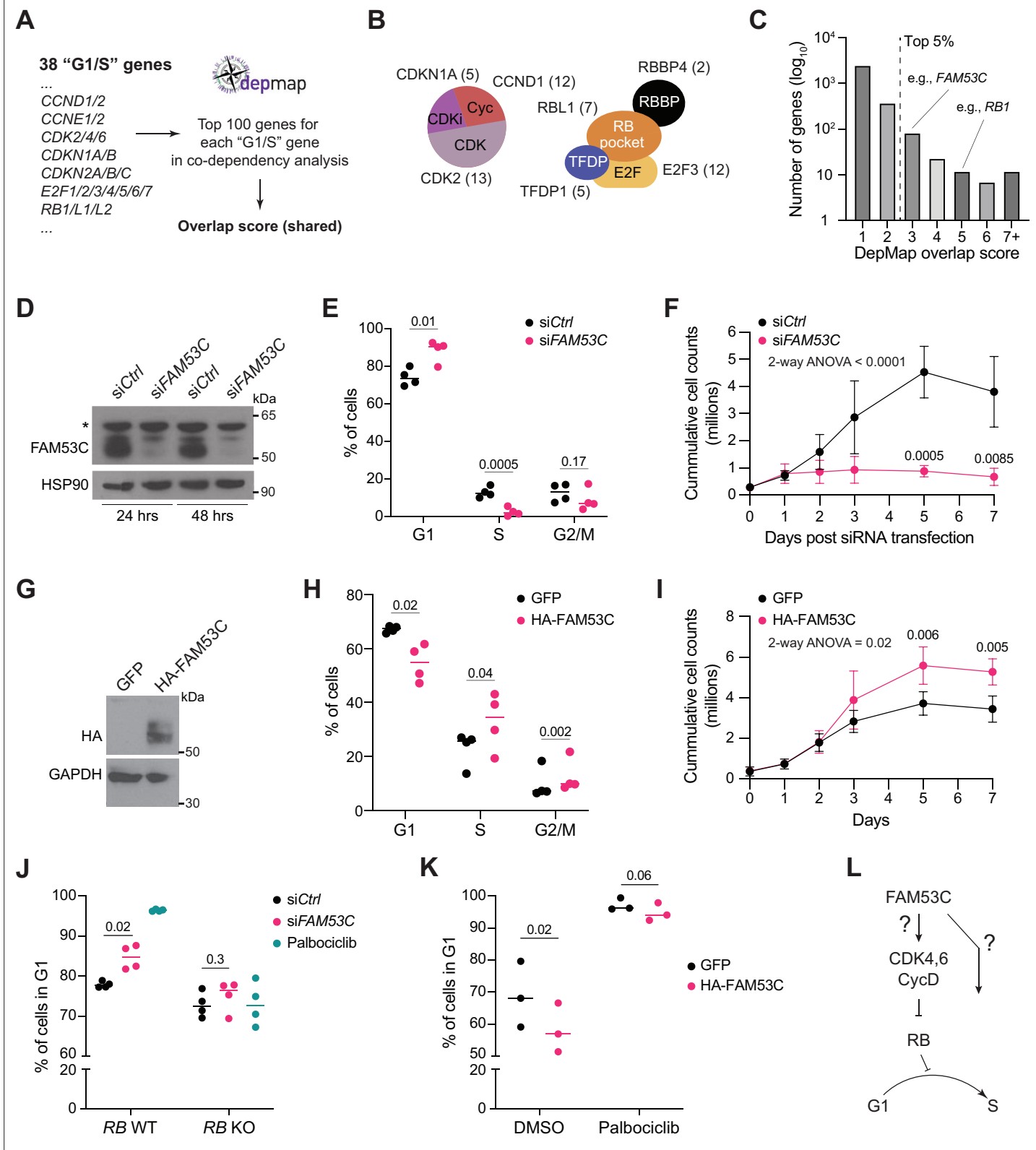

**Figure 1.** Identification of FAM53C as a positive regulator of cell cycle progression in G1. (**A**) Strategy to identify new cell cycle regulators using the DepMap database. See *Supplementary file 1* for the complete list of the 38 genes and associated DepMap data. (**B**) Schematic representation of key factors in the G1/S machinery and scores in the screen for selected factors. CDKi, CDK inhibitors; RBBP, RB binding protein. (**C**) Representation of the number of genes with different overlap scores in the DepMap analysis. *RB1* and *FAM53C* are indicated; the maximal overlap score was 13 for

*Figure 1 continued on next page*

**Figure 1 continued**

the *CDK2* (not shown on graph). The cut-off overlap score value for top candidates was set at 3 (top 5%). See **Supplementary file 2** for complete data. (**D**) Immunoblot analysis for FAM53C in knock-down (si*FAM53C*) RPE-1 cells compared to controls (si*Ctrl*) at 24 hr and 48 hr. HSP90 serves as a loading control. Molecular weights are indicated in kDa. *, non-specific signal. (**E**) Cell cycle analysis by BrdU/PI staining in control and FAM53C knock-down RPE-1 cells (N=4). (**F**) Population growth analysis (cell counts) in control and FAM53C knock-down RPE-1 cells (N=3–5). (**G**) Representative (n=3) immunoblot analysis for FAM53C overexpression (HA-tagged FAM53C) compared to control cells (with GFP expression) in RPE-1 cells. GAPDH serves as a loading control. Molecular weights are indicated in kDa. (**H**) Cell cycle analysis by BrdU/PI staining in control and FAM53C-overexpressing RPE-1 cells (n=4). (**I**) Population growth analysis in control and FAM53C-overexpressing RPE-=1 cells (n=3–5). (**J**) Fraction of cells in G1 from BrdU/PI staining in control (wild-type, WT) and RB knockout (KO) RPE-1 cells, with or without FAM53C knock-down (N=4). (**K**) Fraction of cells in G1 from BrdU/PI staining in control (GFP) and FAM53C-overexpressing RPE-1 cells, with or without palbociclib treatment (N=3). (**L**) Cartoon placing FAM53C in the G1/S transition of the cell cycle. p-Values for (EH), (**J**), and (**K**) were calculated by paired t-test. p-Values for (**F**) and (**I**) were calculated by mixed-model ANOVA followed by post-hoc paired two-tail t-test.

The online version of this article includes the following source data and figure supplement(s) for figure 1:

**Source data 1.** This source data contains source files for panels d-e-f-f-h-i-j-k.

**Source data 2.** This source data file shows the specific areas for the western blots in 1D and 1G.

**Figure supplement 1.** Identification of FAM53C as a positive regulator of cell cycle progression in G1.

candidates; this cutoff score encompassed approximately the top 5% of output values (**Figure 1B and C** and **Supplementary file 2**).

Among the genes with a score greater than or equal to 3 and with no obvious prior direct link to G1 and the RB pathway, we noted *ZZZ3* (score of 6, whose product is a subunit of the Ada-two-A-containing, or ATAC histone acetyltransferase complex **Mi et al., 2018**) and *C1ORF109* (score of 5, whose product is involved in replication stress **Krishnamoorthy et al., 2024**; **Supplementary file 2**). We did not pursue these genes but focused on *FAM53C* (score of 3, Family with sequence similarity 53 C, previously also known as *C5ORF6*) because a similar analysis of DepMap data using a different computational approach also identified this gene as a candidate G1/S regulator (**Wainberg et al., 2021**), and because little is known about FAM53C function.

## FAM53C promotes the G1/S transition of the cell cycle

*FAM53C* has a positive co-dependency correlation with *CCND1* and *CDK4*, and negative correlations with *RB1* in the top 100 co-dependencies for each of the 38 selected factors (**Supplementary file 2**), suggesting that FAM53C may normally act as a promoter of cell cycle progression in G1. Indeed, when we acutely knocked-down FAM53C in immortalized human RPE-1 cells using short interfering RNAs (siRNAs) (**Figure 1D**), we observed a significant accumulation of cells in G1/S and loss of S-phase representation in BrdU/PI assays (**Figure 1E**, **Figure 1—figure supplement 1A and B**). This growth defect was sustained, with no population doublings over 7 days post-transfection (**Figure 1F**). No apoptosis was identified by Annexin V/PI FACS staining, confirming that loss of replication is due to true cell cycle arrest and not cell death (**Figure 1—figure supplement 1C and D**). Knock-down of FAM53C in U2OS osteosarcoma and A549 lung cancer cell lines also led to significant G1 arrest (**Figure 1—figure supplement 1E**). Conversely, upon FAM53C overexpression (**Figure 1G**), we observed a greater fraction of cells in S-phase and increased number of cells at confluency (**Figure 1H and I**, **Figure 1—figure supplement 1F and G**).

Based on the DepMap co-dependency analysis and the knockdown data, we next decided to investigate the pathway interplay between the Cyclin D1/CDK4-RB pathway and FAM53C. We found that knockout of RB in RPE-1 cells significantly abrogated the G1 arrest observed upon FAM53C knockdown; loss of RB also abrogated the effects of the CDK4/6 inhibitor palbociclib in these experiments, as expected (**Figure 1**, **Figure 1—figure supplement 1**). The increased proliferation of RPE-1 cells overexpressing FAM53C could be effectively blocked by palbociclib treatment (**Figure 1K**, **Figure 1—figure supplement 1I**). Altogether, these analyses and functional data identified FAM53C as a regulator of the G1/S transition in immortalized and cancer cells (**Figure 1L**).

## The FAM53C interactome highlights the cell cycle regulatory role of FAM53C

FAM53C remains a largely uncharacterized protein with no known functional domains (**Lai et al., 2000**). To gain further insights into the mechanisms of action of FAM53C in cells and how FAM53C

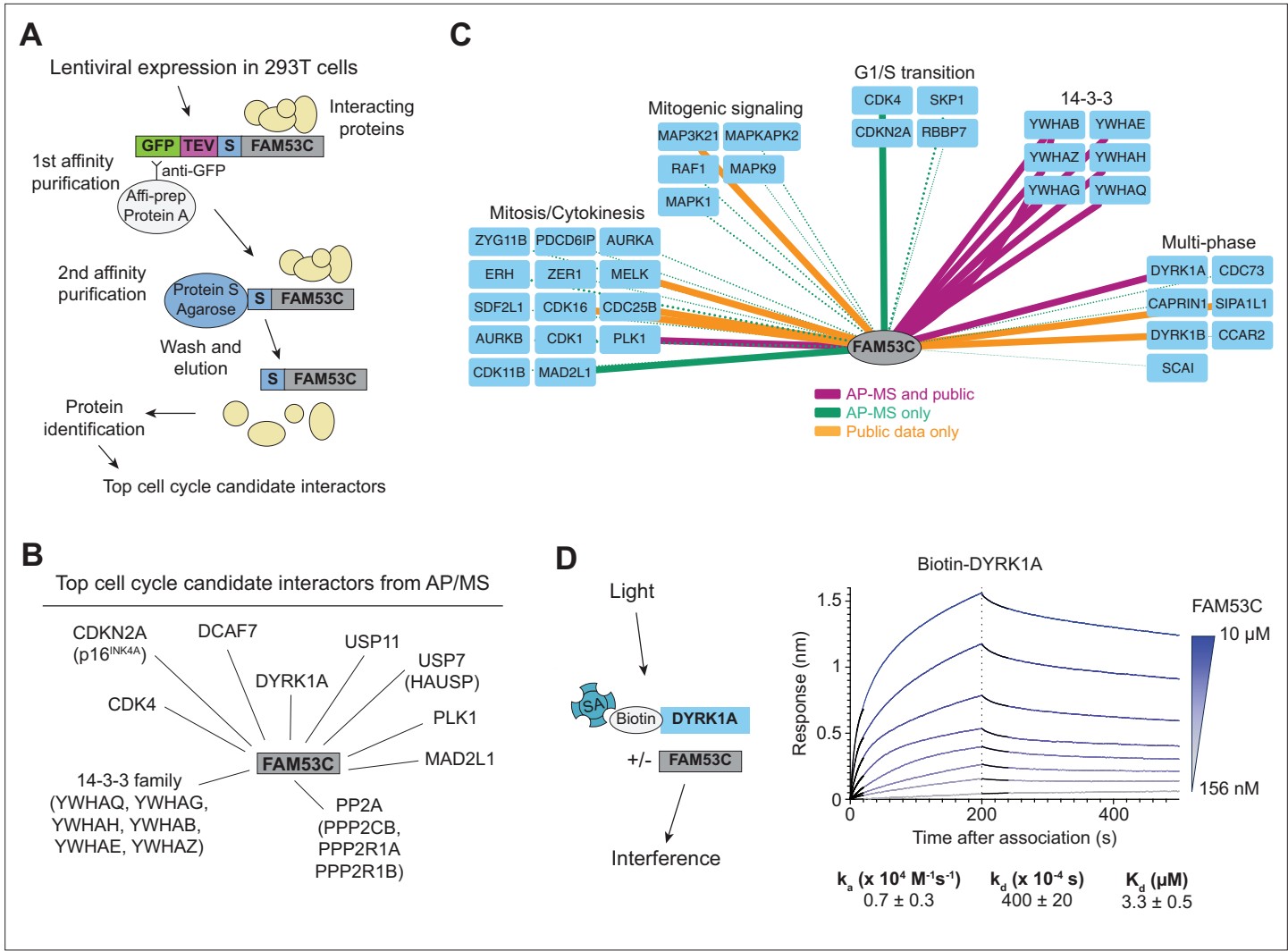

**Figure 2.** The FAM53C interactome identifies cell cycle factors. (**A**) Cartoon representation of the AP/MS experiment to determine the FAM53C interactome. (**B**) Top cell cycle interactors from the AP/MS experiment. See **Supplementary files 3 and 4** for the complete list. (**C**) CytoTRACE analysis integrating the results of the AP-MS experiment with public databases. Solid lines indicate p≤0.05, and line thickness correlates to pNSAF from the AP-MS experiment. (**D**) Biolayer interferometry assay to measure the binding of recombinant FAM53C to DYRK1A-coated streptavidin (SA) sensors. Association begins at 0 s. Dotted line indicates start of dissociation phase. Black solid line indicates portion of the curve that was analyzed for on and off rates. Reported rate constants are from data fitting of observed on and off rates, and the equilibrium constant was determined from the steady-state response analysis.

The online version of this article includes the following source data and figure supplement(s) for figure 2:

**Figure supplement 1.** The FAM53C interactome identifies cell cycle factors.

**Figure supplement 1—source data 1.** This source data shows the uncropped gel with the purified proteins.

**Figure supplement 1—source data 2.** This source data contains the cropped gel with purifed FAM53C.

may connect with the RB pathway, we analyzed the FAM53C interactome in cells. To this end, we expressed a GFP- and S-tagged (localization and affinity purification tag, LAP-tag) lentiviral FAM53C construct in 293T cells, pulled down the FAM53C protein, and performed mass spectrometry on the purified fraction (**Figure 2A**, **Figure 2—figure supplement 1A**). This dual-affinity purification/mass spectrometry (AP-MS) analysis revealed candidate interactors (**Supplementary file 3**), with an enrichment for cell cycle factors (**Supplementary file 4**). Candidate FAM53C interactors in the cell cycle were related to both the G1/S (e.g. CDK4) and the G2/M (e.g. PLK1) transitions, including binding to several subunits of protein phosphatase 2 A (PP2A) and several members of the 14-3-3 family, which may contribute to regulation at multiple cell cycle transitions (**Gardino and Yaffe, 2011**; **Wlodarchak**

*and Xing, 2016*; *Figure 2B* and *Supplementary file 3*). This analysis further points to a role for FAM53C at the G1/S transition of the cell cycle, but also suggests that FAM53C could play some yet unidentified roles at other phases of the cell cycle. While a specific analysis of the FAM53C interactome has not been performed previously, FAM53C was identified in other pull-down experiments. Integrating our AP/MS data with interactome data from public databases confirmed a likely role for FAM53C in the regulation of cell cycle progression, including the G1/S transition (*Figure 2C*).

While FAM53C is likely to have several important partners in cells, DYRK1A stood out as a promising candidate. First, previous proteomic studies focusing on the DYRK1A interactome had identified FAM53C in the list of potential partners of DYRK1A (*Huttlin et al., 2021*; *Menon et al., 2019*; *Roewenstrunk et al., 2019*; *Guard et al., 2019*). Second, our proteomic analysis also identified DCAF7, which is a known functional partner of DYRK1A (*Glenewinkel et al., 2016*; *Xiang et al., 2017*). Third, a recent study indicates a functional relationship between FAM53C and DYRK1A in neurons (*Miyata and Nishida, 2023*). Fourth, strong evidence indicates that DYRK1A kinase activity is a key regulator of the G1/S transition: DYRK1A phosphorylation of LIN52 can promote the activity of the Dimerization Partner, RB-like, E2F, and multivulva class B (DREAM) complex, thereby promoting quiescence (*Litovchick et al., 2011*); DYRK1A phosphorylation of Cyclin D1 at T286 also results in Cyclin D1 degradation, thereby slowing or preventing G1/S progression (*Chen et al., 2013a*; *Soppa et al., 2014*; *Thompson et al., 2015*; *Hille et al., 2016*; *Massey et al., 2021*; *Najas et al., 2015*). Thus, to further assess a potential binding interaction between FAM53C and DYRK1A, we used biolayer interferometry (BLI), an optical biosensing technology that quantifies molecular interactions. After affinity purification of FAM53C and DYRK1A expressed from bacteria, a sortase reaction was used to label the N-terminus of DYRK1A with a biotin tag, which allowed us to efficiently load DYRK1A onto streptavidin-coated BLI sensors. FAM53C association to the sensors was then measured in a dose-dependent manner, and rate and binding constants were determined (*Figure 2D*). Using this approach, we found that FAM53C has an affinity for DYRK1A of $K_d = 3.3 \pm 0.5$ mM, confirming the ability of these two proteins to directly interact. Based on these observations, we decided to pursue the analysis of the interactions between FAM53C and DYRK1A in cells.

## FAM53C acts as an inhibitor of DYRK1A

The direct binding of FAM53C to DYRK1A and the co-dependency scores between FAM53C, DYRK1A, and Cyclin D1 in the DepMap analysis led us to hypothesize that FAM53C may act as a direct inhibitor of DYRK1A kinase activity, in addition to its recently reported role as a regulator of DYRK1A localization in neurons (*Miyata and Nishida, 2023*). To test this idea, we first developed a radio-labeling kinase assay utilizing recombinantly expressed DYRK1A, Cyclin D1, and FAM53C. Cyclin D1 phosphorylation by DYRK1A has been previously observed in vitro at short time points (<30 min; *Chen et al., 2013a*; *Soppa et al., 2014*; *Thompson et al., 2015*; *Hille et al., 2016*; *Massey et al., 2021*; *Najas et al., 2015*), making it a biochemically and biologically relevant substrate to test FAM53C effects on DYRK1A. Upon titrating increasing amounts of FAM53C, we observed a reduction in Cyclin D1 phosphorylation in this assay (*Figure 3A*). We also noted efficient phosphorylation of FAM53C itself in the presence of DYRK1A with lower levels of phosphorylation at lower levels of FAM53C (*Figure 3A*), suggesting that FAM53C may be a competitive substrate and/or an inhibitor of DYRK1A. Phosphorylation of LIN52, another substrate of DYRK1A (*Litovchick et al., 2011*), was also reduced with increasing levels of FAM53C (*Figure 3—figure supplement 1A*). Taken together, these data support a model in which FAM53C can inhibit DYRK1A kinase activity toward key cell-cycle substrates.

DYRK1A dosage has been linked to shifting dynamics between the cell cycle regulators Cyclin D1 and p21, including a cell cycle arrest state characterized by low levels of Cyclin D1 and high levels of p21 associated with increased DYRK1A activity (*Chen et al., 2013a*). When we examined Cyclin D1 and p21 protein levels upon FAM53C knock-down, we found that Cyclin D1 levels decreased while p21 levels increased (*Figure 3B and C*, *Figure 3—figure supplement 1B and C*), consistent with increased DYRK1A activity upon FAM53C loss. Overexpression of FAM53C led to increased levels of Cyclin D1, with no changes in p21 levels (*Figure 3D and E*, *Figure 3—figure supplement 1D and E*). These results further support a model in which FAM53C acts as a DYRK1A inhibitor. These data also raised the possibility that FAM53C levels and p21 levels may not be only linked by DYRK1A activity.

To test whether DYRK1A inhibition could rescue the G1 arrest observed upon *FAM53C* knock-down, we used a highly selective ATP-competitive DYRK1A inhibitor (SM13797 *Jarvis et al., 2023*,

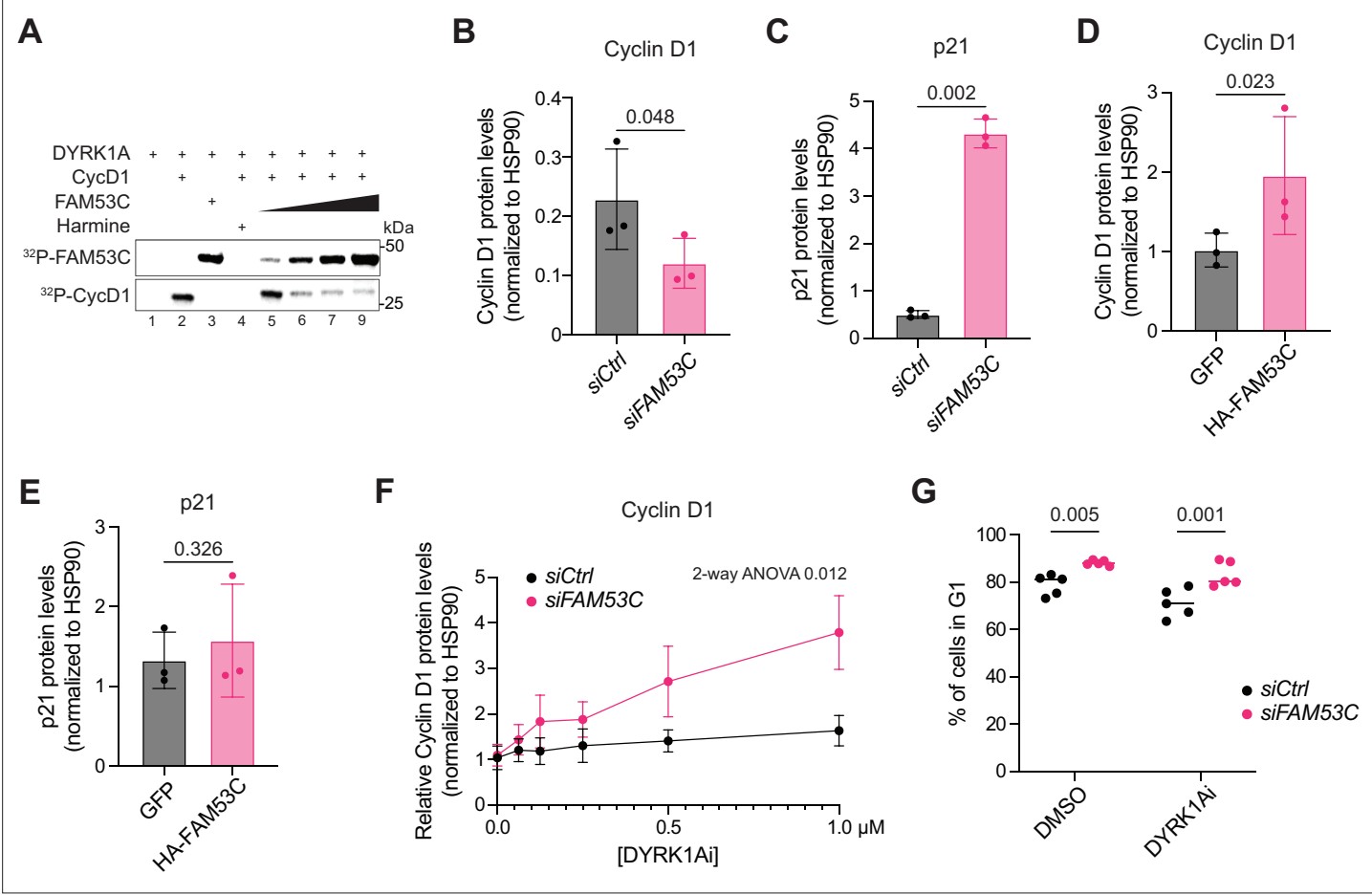

**Figure 3.** FAM53C can inhibit DYRK1A function in cells. (**A**) In vitro phosphorylation assay. Recombinantly expressed Cyclin D1 (1 µM) was phosphorylated by DYRK1A (200 nM) for 15 min alone or in the presence of increasing amounts of FAM53C. FAM53C concentration was 5 µM (lane #3) or 1, 2.5, 5, and 10 µM (lanes #5–8). Lane #1 contains DYRK1A without substrate. The DYRK1A kinase inhibitor Harmine was added at 25 µM. Representative experiment of N=3 experiments. (**B**) Quantification of immunoassays for Cyclin D1 protein levels relative to the loading control HSP90 in FAM53C knock-down (si*FAM53C*) RPE-1 cells compared to controls (si*Ctrl*) at 24 hr (N=3). (**C**) Quantification of immunoassays for p21 protein levels relative to the loading control HSP90 in FAM53C knock-down (si*FAM53C*) RPE-1 cells compared to controls (si*Ctrl*) at 24 hr (N=3). (**D**) Quantification of immunoassays for Cyclin D1 protein levels relative to the loading control HSP90 in RPE-1 cells expressing HA-FAM53C compared to GFP controls (N=3). (**E**) Quantification of immunoassays for p21 protein levels relative to the loading control HSP90 in RPE-1 cells expressing HA-FAM53C compared to GFP controls (N=3). (**F**) Quantification of immunoassays for Cyclin D1 protein levels in control and FAM53C knock-down RPE-1 cells treated with or without different concentrations of the SM13797 DYRK1Ai (N=3 per concentration, 48 hr of knock-down and treatment). (**G**) Fraction of FAM53C knock-down (si*FAM53C*) RPE-1 cells in G1 compared to controls (si*Ctrl*), with or without DYRK1Ai treatment (N=5). p-Values for (**B**), (**C**), (**D**), and (**E**) were calculated by paired t-test. p-Value for (**F**) was calculated by two-way ANOVA.

The online version of this article includes the following source data and figure supplement(s) for figure 3:

**Source data 1.** This source data shows the cropped area for the kinase assay in 3A.

**Source data 2.** This source data shows the cropped area for the kinase assay.

**Figure supplement 1.** FAM53C can inhibit DYRK1A function in cells.

**Figure supplement 1—source data 1.** This source data shows the kinase assay in panel A.

**Figure supplement 1—source data 2.** This source data shows the selected area in the kinase assay.

hereafter DYRK1Ai) (*Figure 3—figure supplement 1F–H*). Notably, the addition of DYRK1Ai to *FAM53C* knock-down RPE-1 cells was more effective at increasing Cyclin D1 levels than in control cells (*Figure 3F*, *Figure 3—figure supplement 1I*), further supporting the model in which FAM53C acts as an active site competitor. However, DYRK1Ai treatment was insufficient to rescue the G1 accumulation induced by FAM53C loss in this context (*Figure 3G*).

Altogether, these experiments place FAM53C as an inhibitor of DYRK1A and a regulator of Cyclin D1 levels. Nevertheless, the lack of rescue of the cell cycle arrest by DYRK1Ai upon FAM53C knock-down suggested that additional mechanisms may be at play in the regulation of cell cycle progression. Notably, we found that p21 levels, which are elevated upon *FAM53C* knock-down in RPE-1 cells, were further elevated in cells treated with increasing doses of DYRK1Ai (*Figure 3—figure supplement 1J and K*). This observation led us to investigate cell cycle arrest mechanisms possibly related to p21 levels downstream of FAM53C knock-down.

## Activation of p53 downstream of FAM53C loss

The gene coding for p21, *CDKN1A*, is known to be regulated at the transcriptional level, including by p53. To determine whether there is transcriptional activation of *CDKN1A* in *FAM53C* knockdown cells, we analyzed the transcriptome of RPE-1 cells 48 hr post-knockdown. Compared to controls, FAM53C knockdown cells exhibited enriched downregulation of genes related to cell cycle processes

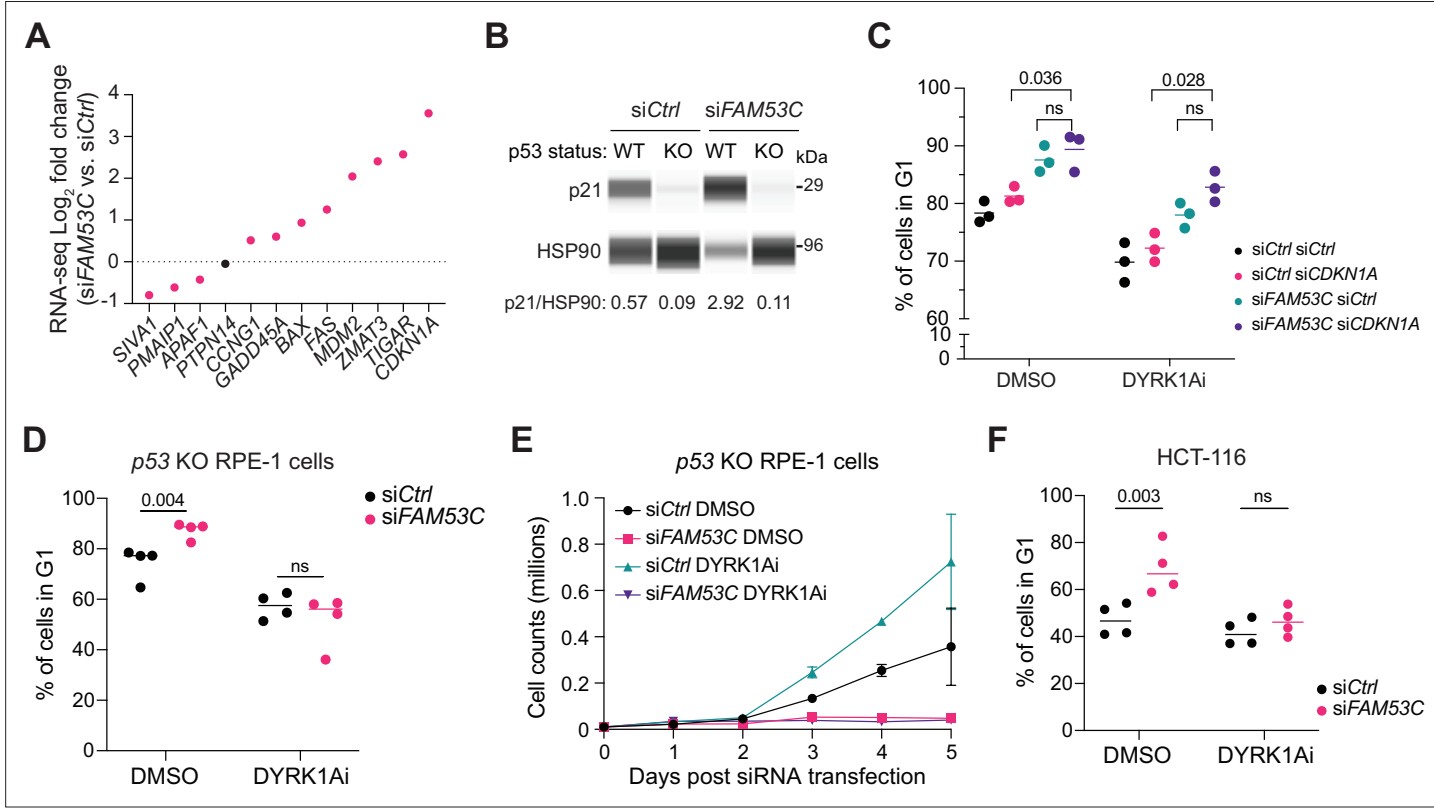

**Figure 4.** FAM53C knock-down activates p53. (**A**) Fold-change analysis of the p53 target genes in FAM53C knock-down cells compared to controls (RNA-seq data). Black dot: p-value >0.05. (**B**) Immunoassay for p21 in *TP53* wild-type (WT) and knockout (KO) RPE-1 cells, with (si*FAM53C*) or without (si*Ctrl*) FAM53C knock-down. HSP90 serves as a loading control (N=1). Values on the bottom represent the ratio between the signal for p21 and the signal for HSP90 in the same lane. (**C**) Fraction of cells in G1 in RPE-1 cells, with or without FAM53C knock-down, with or without p21 knock-down (p21 is encoded by *CDKN2A*), with or without treatment with the DYRK1A inhibitor (DYRK1Ai) (N=3). DMSO is a control for DYRK1Ai. Note that the G1 arrest observed upon FAM53C knock-down is still present in cells with p21 knock-down and DYRK1Ai treatment (ns, not significant). (**D**) Fraction of FAM53C knock-down (si*FAM53C*) *TP53* knockout RPE-1 cells in G1 compared to controls (si*Ctrl*), with or without DYRK1Ai treatment (N=4). Note that the G1 arrest observed upon FAM53C knock-down is absent in cells with *TP53* knockout and DYRK1Ai treatment. (**E**) Cell counts for *TP53* knockout RPE-1 cells, with or without FAM53C knock-down, with or without DYRK1Ai treatment (N=2). Note that cells with FAM53C knock-down and DYRK1Ai treatment do not proliferate despite the decrease in G1 fraction in (**D**). (**F**) Fraction of FAM53C knock-down (si*FAM53C*) HCT-116 cells in G1 compared to controls (si*Ctrl*), with or without DYRK1Ai treatment (N=4). Note that the G1 arrest observed upon FAM53C knock-down is absent in cells with *TP53* knockout and DYRK1Ai treatment. p-Values for (**C**), (**D**), and (**F**) were calculated by paired t-test.

The online version of this article includes the following source data and figure supplement(s) for figure 4:

**Figure supplement 1.** FAM53C knock-down activates p53.

**Figure supplement 1—source data 1.** This source data shows the western blot in panels C and I.

**Figure supplement 1—source data 2.** This source data shows the selected areas in the western blots.

(*Figure 4—figure supplement 1A* and *Supplementary file 5*), along with significant changes in genes coding for key cell cycle factors (*Figure 4—figure supplement 1B*), as would be expected for cells undergoing G1 arrest. Notably, a number of p53 transcriptional targets were significantly upregulated in the knock-down condition, with *CDKN1A* showing the strongest upregulation in the FAM53C knock-down cells (*Figure 4A*) and showing decreased levels in p53 knockout cells (*Figure 4B*). However, p21 knock-down, even with DYRK1Ai treatment, did not rescue the cell cycle arrest observed in FAM53C knock-down RPE-1 cells, indicating that other p53 targets are implicated (*Figure 4C*, *Figure 4—figure supplement 1C and D*).

Based on these observations, we next compared the effects of the FAM53C knockdown on the cell cycle of wild-type and *TP53* knockout RPE-1 cells. Similar to DYRK1A inhibition, ablation of p53 alone was not sufficient to rescue the arrest induced by loss of FAM53C (*Figure 4—figure supplement 1E–H*, see *Figure 3G*). In contrast, the combination of *TP53* knockout and treatment with the DYRK1Ai rescued G1 accumulation in FAM53C knock-down cells (*Figure 4D*, *Figure 4—figure supplement 1E*), indicating that both pathways contribute to this arrest in RPE-1 cells.

We noted that in the G1 rescue condition with DYRK1A inhibition and *TP53* knockout in RPE-1 cells, S-phase values remained largely unchanged while a secondary accumulation of cells in G2/M appeared (*Figure 4—figure supplement 1E*). This observation suggested that release from *FAM53C* knockdown-mediated G1 arrest may result in stress at later phases of the cell cycle in this cell line. Indeed, when we conducted a cell count assay in *TP53* knockout RPE-1 cells, cells treated with control siRNAs grew as expected, but we did not observe increased counts for FAM53C knock-down cells treated with DYRK1Ai over the course of the assay (*Figure 4E*), indicating that these cells, while entering S-phase, are not returning to a normal cell cycle. Cell lysates from cells collected at 48 hr showed an upregulation of cleaved caspase 3 (CC3) only in DYRK1Ai-treated FAM53C knock-down cells lacking p53 (*Figure 4—figure supplement 1I*), suggesting that bypass of FAM53C-loss arrest may lead to significant cell stress in later stages of the cell cycle and death.

In the DepMap dataset, p53 restricts the expansion of RPE-1-derived cell lines, and Cyclin D1 is critical for their expansion. We also found that HCT-116 colon cancer cells (which are p53 wild-type) are dependent on Cyclin D1 for their expansion, similar to RPE-1 cells, but largely independent on p53 (*Figure 4—figure supplement 1J*). Notably, FAM53C knock-down in these cells led to G1 arrest, which was rescued by treatment with the DYRK1A inhibitor (*Figure 4F*, *Figure 4—figure supplement 1K*), indicating that DYRK1A is a critical mediator of cell cycle arrest in HCT-116 cells upon FAM53C knock-down and that loss of p53 is not always required to rescue the G1 arrest observed upon FAM53C knock-down. We did not examine the long-term proliferative potential of these cells, but DYRK1A inhibition has been previously shown to negatively affect the G2/M program in these cells (*Laham et al., 2024*). Overall, these experiments identify activation of the p53 pathway and other stress signals downstream of FAM53C loss, in addition to activation of DYRK1A (*Figure 4—figure supplement 1L*).

## Consequences of *FAM53C* inactivation in human cortical organoids in culture

Based on the dosage-dependent role of DYRK1A in brain development, we wondered if loss of FAM53C, which can lead to DYRK1A kinase activation, may result in brain developmental phenotypes. As a first way to test this idea, we knocked out *FAM53C* in human induced pluripotent stem cells (iPSCs) using CRISPR/Cas9 (*Figure 5A*, *Figure 5—figure supplement 1A-C*). The *FAM53C* knockout in iPSCs was compatible with their survival and their stemness (*Figure 5—figure supplement 1D and E*), which allowed us to differentiate control and knockout cells into human cortical organoids (hCOs). After 25 days of differentiation, we noted that FAM53C mutant organoids were smaller in size compared to controls at this stage (*Figure 5B and C*). Using EdU incorporation assays, we also observed decreased proliferation in the *FAM53C* knockout hCOs compared to controls (*Figure 5D*, *Figure 5—figure supplement 1F*). Loss of FAM53C in hCOs did not affect DYRK1A levels, but the ratio between phosphorylated Cyclin and total Cyclin D1 levels was greater in knockout hCOs compared to controls, and p21 levels were elevated (*Figure 5E–H*). While a complete analysis of the role of FAM53C in the development of various cell types and structures in hCOs remains to be performed, these observations further support a role for FAM53C in the control of cell cycle progression in G1 in a neural development context.

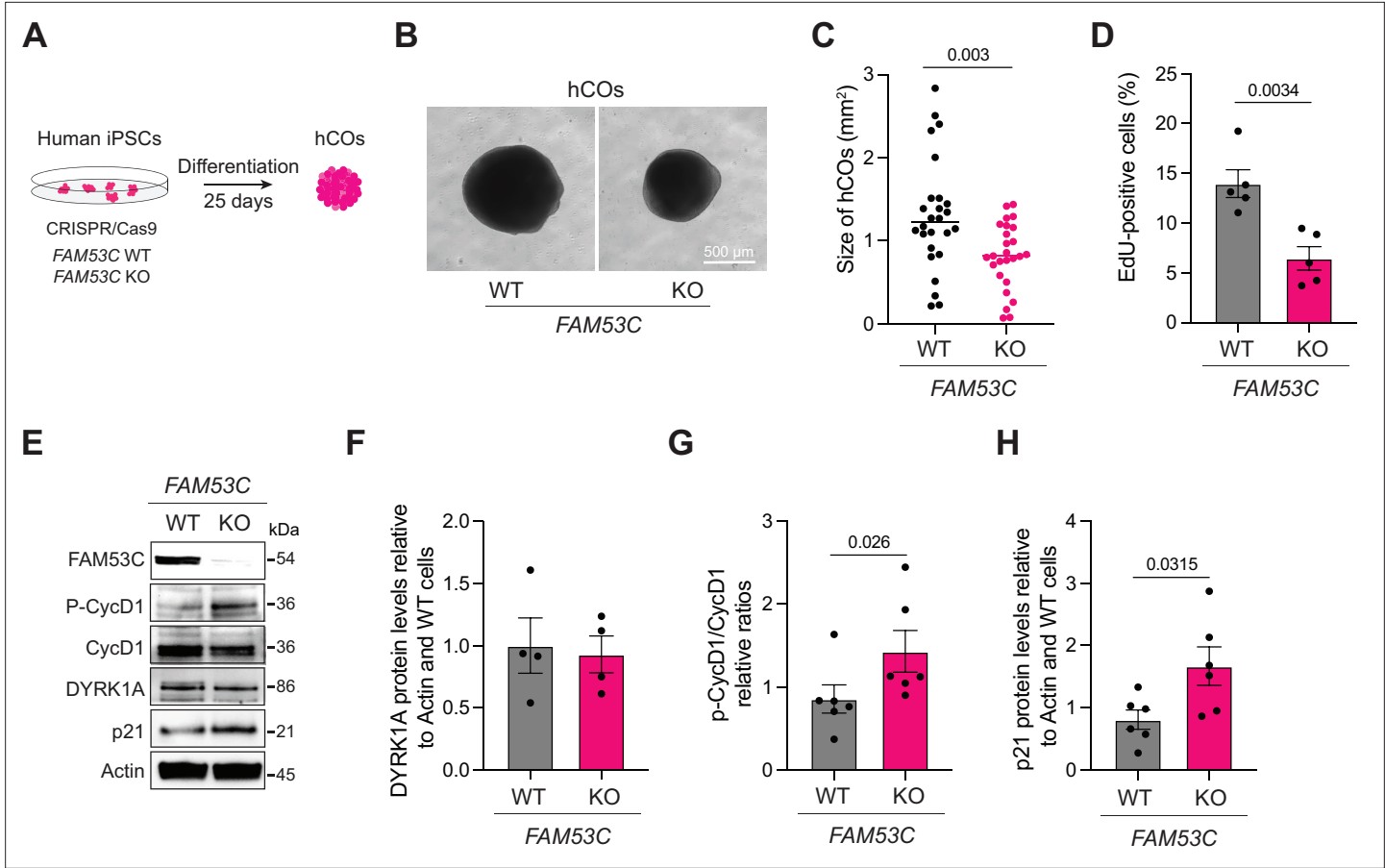

**Figure 5.** Loss of FAM53C impairs the development of human cortical organoids. (**A**) Cartoon of the differentiation protocol from human induced pluripotent stem cells (iPSCs) to human cortical organoids (hCOs). (**B**) Representative images of wild-type and knockout hCOs. Scale bar, 500 μm. (**C**) Quantification of (**B**). (**D**) Quantification of Edu-positive cells in wild-type and knockout hCOs. (**E**) Representative immunoblot analysis of wild-type (WT) and *FAM53C* knockout (KO) hCOs. β-actin serves as a loading control. (**F**) Quantification of (**E**) for DYRK1A relative to β-actin. (**G**) Quantification of (**E**) for phospho-Cyclin D1 relative to Cyclin D1 levels. (**H**) Quantification of (**E**) for p21 levels relative to β-actin levels. p-Values for (**C**), (**D**), (**F**), (**G**), and (**H**) calculated by paired t-test.

The online version of this article includes the following source data and figure supplement(s) for figure 5:

**Source data 1.** This source data shows the western blots in panel E.

**Source data 2.** This source data shows the selected areas for the western blots.

**Figure supplement 1.** Loss of FAM53C impairs the development of human cortical organoids.

## *Fam53C* knockout mice are viable and display only mild phenotypes

These data, and developmental roles for DYRK1A also in the brain and outside of the brain (*Thompson et al., 2015*; *Hille et al., 2016*; *van Bon et al., 1993*) led us to investigate the possible phenotypes of *Fam53C* knockout mice. We obtained mice with a deletion of *Fam53C* exon 4 from the International Mouse Phenotyping Consortium (IPMC; *Groza et al., 2023*; *Figure 6A*). *Fam53C$^{-/-}$* and *Fam53C$^{+/-}$* mice were recovered at the expected frequency from *Fam53C$^{+/-}$*crosses in the IMPC colony (with a low number of mice analyzed, *Figure 6—figure supplement 1A*), but with a significant trend towards fewer of the homozygous mutant mice in our colony at Stanford University (with more mice analyzed, *Figure 6B*). Based on the cell cycle arrest phenotypes observed in culture, *Fam53C$^{-/-}$* may have been expected to have a reduced body size, similar for example to Cyclin D1 knockout mice (*Sicinski et al., 1995*; *Fantl et al., 1995*). We observed a significantly lower body mass at weaning for males in our colony (*Figure 6C*). This phenotype may be dependent on the husbandry conditions, as there were no significant differences in body weight between knockouts and wild-type controls, males or females, in the IMPC dataset (although again with only n=8 mice in each group there; *Figure 6—figure supplement 1B*). Histological analysis of the brain and other tissues and organs in adult control and knockout

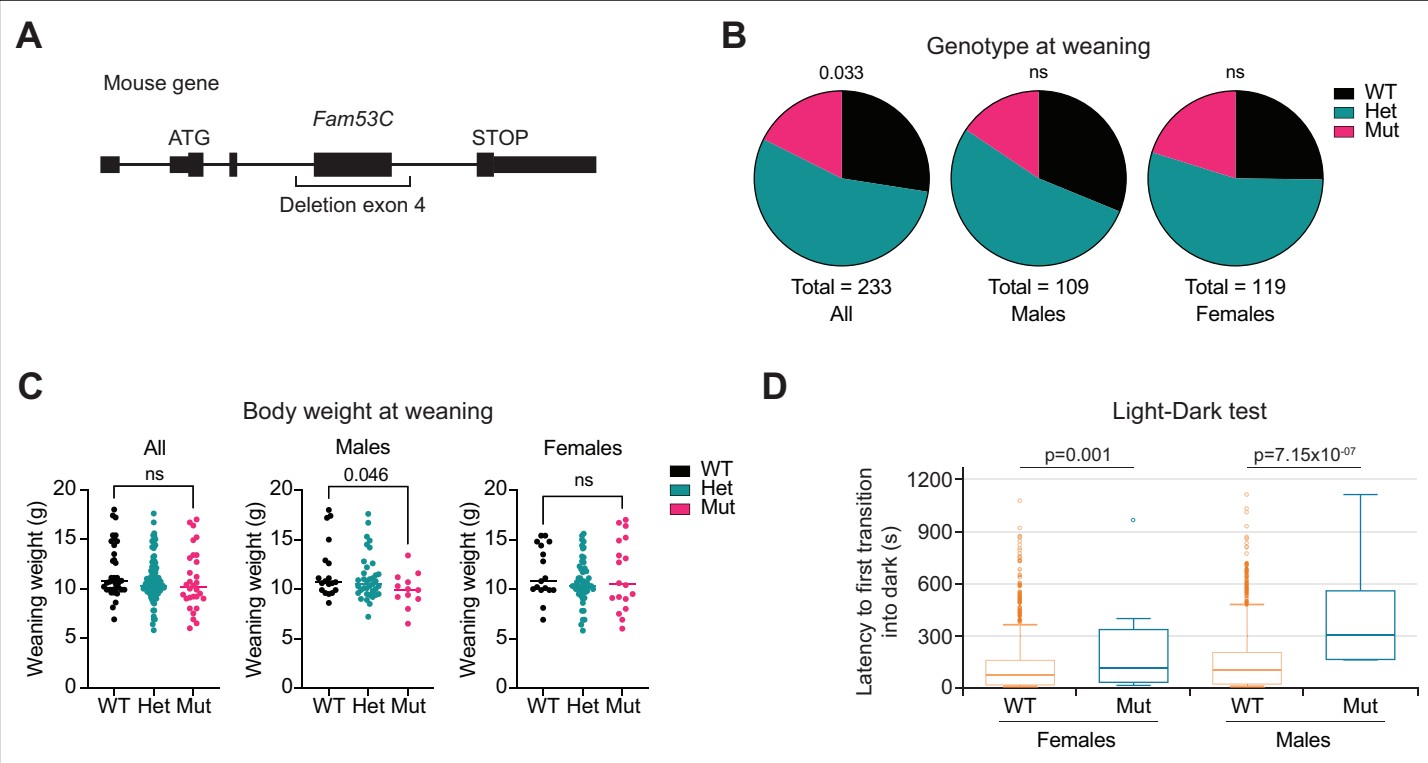

**Figure 6.** *Fam53C* knockout mice are viable and display limited phenotypes. (**A**) Cartoon of the *Fam53C* mutant allele, with deletion of the major coding exon (exon 4; not to scale). (**B**) Genotypes of mouse pups at weaning from *Fam53C*$^{+/-}$crosses at the Stanford facility. WT: wild-type; Het: heterozygous mutant mice; Mut: homozygous mutant mice. (**C**) Body weight analysis at weaning for mice generated by *Fam53C*$^{+/-}$crosses. (**D**) Measure of the latency to first transition into a dark chamber (Light-Dark test) for control (n=1998 males and n=2037 females – including historical controls) and *Fam53C* knockout mice (n=6 males and n=7 females). Note the extremely different sizes in the cohorts of control and mutant mice, which may affect the results of the statistical analysis. p-Values for (**B**) calculated by a Chi-squared test. p-Values for (**C**) calculated by unpaired t tests. The details of the Linear Mixed Model framework for statistical analysis for (**D**) are available on the IMPC website.

The online version of this article includes the following figure supplement(s) for figure 6:

**Figure supplement 1.** *Fam53C* knockout mice are viable and display limited phenotypes.

mice did not reveal any gross defects (*Figure 6—figure supplement 1C*). We note that the IMPC behavioral analysis had shown significant differences between wild-type and *Fam53C*$^{-/-}$ male mice in a 'latency to first transition into dark' test, in which the knockout mice showed a decreased exploration of a new environment, suggestive of possibly increased anxiety (n=6 *Fam53C*$^{-/-}$ males and n=7 *Fam53C*$^{-/-}$ females compared to historical controls; *Figure 6D*). However, we did not repeat similar experiments in our colony. In a survival study, we did not detect any differences between controls and *Fam53C*$^{-/-}$ mice (*Figure 6—figure supplement 1D*). These observations in mice with minimal phenotypes linked to FAM53C loss are in stark contrast to the strong phenotypes observed in cells in culture and suggest mechanisms of compensation in vivo that remain to be identified.

## Discussion

Here, we sought to identify novel regulators of the G1/S transition using the DepMap database as a starting point. Our analysis led us to focus on FAM53C, a factor that was previously uncharacterized as a cell cycle factor. Our data place FAM53C upstream of the CycD/CDK4,6-RB and p53-p21 pathways at the G1/S transition of the cell cycle. Our data further show that DYRK1A is a key partner of FAM53C in the control of cell cycle progression at this transition.

FAM53C is conserved throughout vertebrates, but little is known about the function of the FAM53 protein family. Previous work on FAM53B in medaka fish had shown a role in linking DNA-binding proteins to modulators of transcription in the control of proliferation (*Thermes et al., 2006*). Previous

work on mammalian FAM53A had also suggested a connection with Bone Morphogenetic Protein/Transforming Growth Factor beta (BMP/TGFβ) signaling (*Bergemann et al., 2008*) and with p53 signaling (*Zhang et al., 2019*), and our data connecting FAM53C loss to p53 activation may suggest an ancestral connection to p53 for this family.

DYRK1A controls the G1/S transition of the cell cycle (*Chen et al., 2013a*; *Soppa et al., 2014*; *Thompson et al., 2015*; *Litovchick et al., 2011*; *Massey et al., 2021*; *Najas et al., 2015*; *Tschöp et al., 2011*; *Kruitwagen et al., 2018*), but also plays a role in differentiation, especially in developing neurons (*Ryoo et al., 2007*; *Laguna et al., 2013*; *Méjécase et al., 2021*; *Wegiel et al., 2004*; *Göckler et al., 2009*; *Neumann et al., 2018*). Thus, it is possible that FAM53C contributes to the control of cellular differentiation, including by regulating the cellular localization of DYRK1A (*Miyata and Nishida, 2023*), its kinase activity, or by other mechanisms. Because of the role of DYRK1A in the developing brain and in defects associated with Down syndrome (*de Souza et al., 2023*; *Wegiel et al., 2011*; *Ryoo et al., 2007*), changes in FAM53C levels and/or activity may be expected to affect brain development. The behavioral data from the International Mouse Phenotyping Consortium (IMPC) support this idea, even though these experiments were performed with a small number of animals and will require future validation. In particular, the cellular basis of this behavioral phenotype, if repeated, will require a more in-depth analysis of the brain of mutant mice, possibly at different time points during development, as our histological analysis did not reveal any major changes in the brain of knockout mice compared to controls in adult animals. Notably, human *FAM53C* variants in public databases are linked to phenotypes such as height and Alzheimer's disease ('co-localization analysis' from the Open Targets Genetics initiative *Ghoussaini et al., 2021*), as well as IGF1 serum levels (data from the Global Biobank Engine *McInnes et al., 2019*). Disruption of IGF1 signaling is a candidate contributor to stunted growth and brain development defects in Down syndrome (*Araya et al., 2022*). These observations provide further rationale to investigate FAM53C function ex vivo and in vivo, both in the regulation of cell cycle progression and in brain development and function.

Our data indicate that FAM53C controls DYRK1A kinase activity beyond its previously described ability to control DYRK1A cellular localization (*Miyata and Nishida, 2023*). Our phosphorylation assays further suggest that DYRK1A may phosphorylate FAM53C, suggesting a role for FAM53C as a possible competitive inhibitor against other substrates. But it is also possible that this phosphorylation in turn affects FAM53C localization and/or activity. The identification and mutation of the DYRK1A phosphorylation site(s) on FAM53C will be a first step to address this point in the future.

While our work identifies a FAM53C-DYRK1A-CycD/CDK4,6-RB pathway at the G1/S transition of the cell cycle, it is likely that FAM53C has other functions in cells. In particular, our AP-MS interactome data suggest that FAM53C may play additional roles later in the cell cycle, including during G2/M through associations with other kinases like PLK1 and AURKA. Whether these kinase interactions are inhibitory, similar to the FAM53C-DYRK1A interaction, remains to be determined. We did not observe cell cycle arrest at other stages of the cell cycle beyond G1, but future experiments, including in synchronized cell populations or by single cell tracking, will help address the role of FAM53C at other key cell cycle checkpoints.

With the approval of CDK4/6 inhibitors for the treatment of breast cancer and the development of a variety of CDK inhibitors (*Sherr et al., 2016*; *Suski et al., 2021*; *Goel et al., 2022*), it has become even more important than before to better understand the mechanisms regulating the activity of these kinases. Our data indicate that low levels of FAM53C may activate DYRK1A kinase activity towards Cyclin D, thereby decreasing CDK4/6 activity in cells. Our mass spectrometry analysis also identified CDK4 as a possible interactor of FAM53C. Although we did not pursue this observation, a possible direct interaction between FAM53C and CDK4 may also regulate CDK4 activity in cells. The identification of FAM53C as a regulator of the DYRK1A-CycD/CDK4,6-RB pathway suggests that strategies to decrease FAM53C levels in cells may help enhance the anti-tumor effects of CDK inhibitors.

## Methods
### Animal studies

We imported *Fam53C* knockout mice from the International Mouse Phenotyping Consortium (IMPC *Groza et al., 2023*, https://www.mousephenotype.org/) (*Fam53cem1(IMPC)J* allele, C57BL/6NJ background). This allele was generated by electroporating the Cas9 protein along with 2 guide

sequences 5'-ACTGCATTTTTGAGGAGGAG-3' and 5'-GTTAAAATTCAATACTGCCA-3', which resulted in a 1151 bp deletion beginning at Chromosome 18 position 34,767,928 bp and ending after 34,769,078 bp (GRCm38/mm10). This mutation deletes exon 4 and 366 bp of flanking intronic sequence including the splice acceptor and donor and is predicted to cause a change of amino acid sequence after residue 46 and early truncation 27 amino acids later. Genotyping was performed using the Mouse Direct PCR kit (for genotyping, B40015) following the protocol provided by the Jackson Laboratory (strain Stock No: 032892, C57BL/6NJ-*Fam53c*$^{em1(IMPC)J/Mmjax}$) using a forward primer for the wild-type allele (5'-CCTGGGAACTCTTCTGTCTAGAGT-3'), a forward primer for the mutant allele (5'-TGGCATTACCACTTCACAGC-3'), and a common reverse primer (5'-CTAAGGACTAACTTGACAGG GCAGA-3') (wild-type band: 99 bp; mutant band: 90 bp). We note that we have not been able to detect the FAM53C protein reliably with current antibodies in immunoblot on mouse cells, making it impossible to ascertain full knockout at the protein level.

Data generated by the IPMC were downloaded from their website (https://www.mousephenotype.org/data/genes/MGI:1913556; *Groza et al., 2023*). For the cohort generated at Stanford to measure body weight and to assess phenotypes during aging, mice were generated from heterozygous crosses.

Histological analysis was performed on hematoxylin and eosin-stained paraffin sections prepared by HistoWiz.

## DepMap analysis

The DepMap analysis was performed directly from the DepMap portal in 2020 (20q2 datasets; https://depmap.org/portal; *Tsherniak et al., 2017*).

## Cell culture and small molecules

Human cell lines were grown in DMEM high glucose medium supplemented with 10% bovine growth serum (BGS; Thermo Fisher Scientific) and 1% penicillin-streptomycin-glutamine (Gibco). All cell lines tested negative for mycoplasma and were verified by STR profiling. RB and p53 mutant RPE-1 cells were a kind gift from Evgeny Zatulovskiy and Jan Skotheim (*Zatulovskiy et al., 2020*). Control and p53 knockout HCT116 cells were a kind gift from Mengxiong Wang and Laura Attardi.

The following small molecule inhibitors were used in cells: palbociclib HCl (Selleckchem #S1116) and puromycin (Thermo Fisher #A1113803). For DYRK1A kinase inhibition, we used SM13797 from Biosplice (*Jarvis et al., 2023*) at a concentration of 1 µM with volume-matched DMSO vehicle control unless otherwise noted.

Human induced pluripotent stem cells (hiPSCs) were cultured in Essential 8 medium (Thermo Fisher Scientific, A1517001) on cell culture plates coated with Vitronectin (Thermo Fisher Scientific, 14190). They were passaged approximately every 6 days using 0.5 mM EDTA (Thermo Fisher Scientific, 15575) once they reached about 80% confluency. Cell integrity was confirmed using high-density SNP arrays, and routine PCR testing was performed to ensure cultures remained free of mycoplasma.

The differentiation of iPSCs into the three germ layers was performed using the STEMdiff Trilineage Differentiation Kit (STEMCELL Technologies, #05230).

## Knock-down, knock-out, and overexpression

FAM53C knock-down experiments in human cell lines were performed using 5 µM of ON-TARGETplus pooled siRNAs (Horizon – Dharmacon), and ON-TARGET Control #1 as negative controls. Cells were transfected using the Lipofectamine RNAiMAX reagent, following the manufacturer's recommendations. Lentiviral vectors encoding HA-FAM53C (C-terminal tag; VectorBuilder) or GFP control (a gift from the Artandi lab). Lentivirus was produced in 293T cells, and cells were infected with viral supernatant on 2 consecutive days, followed by 4 days of selection with puromycin.

hiPSCs ($8 \times 10^5$ cells) were electroporated in P3 primary cell nucleofector solution (Lonza, V4XP-3032) with 2 µg of FAM53C KO Plasmid containing GFP expression system (Santa Cruz Biotechnology, sc-412179) using a 4D-Nucleofector (program: DC100; Lonza). After nucleofection, the cells were transferred to Matrigel-coated plates (Corning, 354277) and cultured in mTESR medium (STEMCELL Technologies, 100–0276) with 10 µM ROCK inhibitor (Y27632) at 37 °C for 24 hr. GFP-positive cells were sorted using a FACSAria II flow cytometer (BD Biosciences) and cultured immediately in mTESR medium containing 10 µM Y27632 and Antibiotic-Antimycotic (Thermo Fisher Scientific, 15240062). Approximately 300 GFP-positive cells were plated per well in Matrigel-coated six-well plates. Colonies

were harvested in QuickExtract DNA Extraction Solution (LGC Biosearch Technologies, QE09050). Then, the genomic region around the CRISPR-Cas9 target site for *FAM53C* was amplified by PCR using Q5 Hot Start High-Fidelity 2 X Master Mix (New England Biolabs, M0494S). To select the *FAM53C* knockout clone, we used the following primer pair: forward, 5′-GGCCCAAGATTCCTCTCGAC-3′, and reverse, 5′-GGGTCTCACCCCAGTTTCTC-3′. To examine the specific region of the knockout, RNA was extracted from the initially selected clones using the RNeasy Plus Mini Kit (Qiagen, 74136). cDNA synthesis was then performed using the SuperScript IV CellsDirect cDNA Synthesis Kit (Invitrogen, 11750150). The cDNA was amplified by PCR and subsequently sequenced. The primers used for this process were: forward, 5′-GCAGACTCTGGATGAGCTGAAATG-3′, and reverse, 5′-TCTCTTGGTGGCAGGGATACT-3′. The sequencing results confirmed that the clone had a deletion of exon 3 and exon 4 (844 bp) (see gene map in *Figure 5—figure supplement 1B and C*). Potential off-target effects of the sgRNA were assessed using the RGEN tool (https://www.rgenome.net/), with PCR primers for detecting mutations at potential off-target sites (*ZFAND5* forward, 5′-GCGGCGAGTGCGTTAGT-3′, and reverse, 5′-TTTGTTTCTCTGGGTCGTGGTG-3′), *MAP6D1* (forward, 5′-GGCTACTCGGACCTCGACA-3′, and reverse, 5′-GGTTCCAACTCGGCTGAAGG-3′), and *DCAF10* (forward, 5′-AAGTTTGGGTCAAGATCCTGGT-3′, and reverse, 5′- AAGGCCAAGTATACTCATAAGTGAGG –3′) (see sequencing results in *Figure 5—figure supplement 1B and C*).

## Generation of hCOs from hiPSCs

The generation of human cortical organoids (hCOs) from hiPSCs followed a previously published protocol (*Yoon et al., 2019*). In brief, hiPSCs were dissociated into single cells using Accutase (Innovate Cell Technologies, AT-104) and seeded into AggreWell plates (STEMCELL Technologies, 34815) at a density of $3\times10^6$ cells per well, using Essential 8 medium (Thermo Fisher Scientific, A1517001) supplemented with 10 µM Rock inhibitor Y-27632 (Selleckchem, S1049). The next day, spheres were collected and transferred to 10 cm ultra-low-attachment dishes (Corning, 3262). For the first 6 days, they were cultured in Essential 6 medium (Thermo Fisher Scientific, A1516401) with SB-431542 (10 µM, Tocris, 1614), dorsomorphin (2.5 µM Sigma-Aldrich, P5499), and XAV-939 (2.5 µM, Tocris, 3748), with daily medium changes. From Day 7, hCOs were maintained in Neurobasal A medium (Thermo Fisher Scientific, 10888) containing B27 (Thermo Fisher Scientific, 12587), along with 20 ng/ml epidermal growth factor (R&D Systems, 236-EG) and 20 ng/ml basic fibroblast growth factor (R&D Systems, 233-FB), with daily medium changes for 8 days, followed by changes every other day until Day 25. At day 25, the hCOs were switched to a Neurobasal A medium with B27, 20 ng/ml brain-derived neurotrophic factor (BDNF; Peprotech, 450–02), and 20 ng/ml NT3 (Peprotech, 450–03) for an additional 20 days, with medium changes every other day. From Day 45 onward, the hCOs were maintained in Neurobasal A medium with B27, with medium changes every 3–4 days.

## Quantitative immunoassay, immunoblot analysis, and immunofluorescence

For immunoassays with human cell lines, cell extracts were prepared in RIPA buffer with Roche cOmplete ULTRA proteasome and PhosSTOP phosphatase inhibitor cocktail tablets (Millipore Sigma #05892791001 and PHOSS-RO). Protein extracts were quantified using the Pierce BCA Protein Assay Kit according to the manufacturer's instructions (Thermo Fisher Scientific, 23227). For quantitative immunoassays, the capillary-based Simple Western assay was performed on the Wes system (ProteinSimple) according to the manufacturer's protocol with 1 µg of protein used per lane. Compass software (ProteinSimple) was used for protein quantification, using the default settings unless otherwise specified. For immunoblot analysis, cell lysates were denatured in Laemmli buffer and boiled at 95 °C for 5 min. 10 µg of protein was loaded into each lane unless otherwise specified. Samples were run on NuPAGE 4 to 12% Bis-Tris gels (Thermo Fisher Scientific # NP0322BOX) and transferred to nitrocellulose membranes using iBlot 2 transfer stacks (Thermo Fisher Scientific IB23002). Membranes were blocked in 10% milk in TBS-T (20 mM Tris, 150 mM NaCl, 0.1% Tween 20) for 1 hr. Primary antibodies were diluted in 5% milk in TBS-T and incubated overnight at 4 °C. Secondary antibodies (Jackson ImmunoResearch anti-rabbit #111-035-144; anti-mouse #115-035-003) were diluted 1:10,000 in 5% milk in TBS-T and incubated for 1 hr at room temperature. Chemiluminescence was detected using Amersham ECL Prime Western Blotting Detection Reagent (GE Healthcare #RPN2236).

The following primary antibodies were used for human cell lines: HSP90 (1:2000; Cell Signaling Technology, #4877), GAPDH (1:5000; Thermo Fisher, PA5-79289), FAM53C (1:250; Thermo Fisher Scientific, PA5-60125), HA tag (1:1000; Cell Signaling Technology, #3714), p21 (1:1000; Cell Signaling Technology, #2947), Cyclin D1 (1:500; Cell Signaling Technology, #2922), and DYRK1A (1:200; Cell Signaling Technology, #8765).

For experiments with hCOs, the organoids were lysed on ice using RIPA buffer with protease (Santa Cruz Biotechnology, sc-24948A) and phosphatase inhibitors (GenDEPOT, P3200-001) through gentle agitation, and incubated at 4 °C for 1 hr. The lysates were then centrifuged at 14,000 × *g* for 15 min, and the supernatant was collected. Protein concentrations in the supernatant were measured using the Pierce BCA Protein Assay Kit (Thermo Fisher Scientific, 23225). The lysates were denatured with Bolt LDS Sample Buffer (Invitrogen, B0007) at 95 °C for 5 min. Then, 7 µg of the samples were loaded onto Bolt 4–12% Bis-Tris Protein Gels (Invitrogen, NW04120BOX) and transferred to a PVDF membrane using the iBlot 2 Transfer Stacks and iBlot2 system (Method: P0; Thermo Fisher Scientific, IB24002). The membranes were blocked with 5% BSA (GenDEPOT, A0100-005) in TBST (Tris-buffered saline with 0.1% Tween 20, Boston BioProducts, BM-301X) for 1 hr. They were then incubated overnight at 4 °C with primary antibodies in TBST containing 5% BSA: DYRK1A (1:1000; Abcam, ab65220), FAM53C (1:1000; Thermo Fisher Scientific, PIPA5114093), Cyclin D1 (1:1000; Cell Signaling, #2922 S), p-Cyclin D1 (1:1000; Cell Signaling, #3300), P21 (1:1000; Cell Signaling, #2947 S), and β-actin (1:1000; Cell Signaling, #4970 S). After washing with 5% TBST, the membranes were incubated with horse-radish peroxidase-conjugated secondary antibodies at room temperature for 1 hr: Anti-mouse IgG (1:2000; Cell Signaling, #7076), and Anti-rabbit IgG (1:2000; Cell Signaling, #7074). The SuperSignal West Femto Maximum Sensitivity Substrate (Thermo Fisher Scientific, 34095) was used for signal development, and the iBright 1500 (Thermo Fisher Scientific, A44114) was used for protein band detection. Band intensities were quantified using ImageJ software (version 1.53t, NIMH, Bethesda, MD) with normalization to background and to the β-actin control.

For immunofluorescence analysis of differentiated iPSCs, cells were fixed in a 10% formalin solution (Sigma) at 4 °C for 15 min, followed by washing with PBST (PBS with 0.1% Tween 20). Permeabilization was carried out using PBS containing 0.1% Triton X-100 (Sigma) for 20 min. Then, the samples were incubated in 3% BSA for 1 hr before being treated overnight at 4 °C with the respective primary antibodies: PAX6 for ectoderm (1:300; BioLegend, PRB-278P), Brachyury for mesoderm (1:300; R&D Systems, AF2085-SP), and SOX17 for endoderm (1:300; Cell Signaling, 81778 S). The cells were then rinsed with PBST and incubated with Alexa Fluor-conjugated secondary antibodies (1:500; 488 or 594; Thermo Fisher Scientific) for 1 hr at room temperature. Samples were observed using a Zeiss LSM 980 confocal microscope (Carl Zeiss).

## Cell cycle and cell death analyses

Cells were counted using a Countess 3 instrument on default settings (Thermo Fisher Scientific), and dead cells were excluded using 0.4% Trypan Blue solution (Thermo Fisher Scientific, T10282). For palbociclib treatment (with DMSO as a control), cells were treated with 0.5 µM for 24 hr. Cell cycle analysis was performed using BrdU pulsed for 2 hr (10 µg/mL, Calbiochem #203806) and propidium iodide (PI) staining (*Chaikovsky et al., 2021*) unless otherwise indicated. For cell death assays, Annexin V/PI staining was performed and analyzed as described before (*Chaikovsky et al., 2021*). Flow cytometry was performed on a BD FACSAria II (BD Biosciences), and data was collected using BD FACSDiva. Data were analyzed using Cytobank Community software (Beckman Coulter Life Sciences).

For experiments with hCOs, the EdU assays were performed using the Click-iT EdU Alexa Fluor 647 Flow Cytometry Assay Kit (Thermo Fisher Scientific, C10424), following the manufacturer's instructions. Briefly, hCOs were incubated with 10 µM EdU for 24 hr, then dissociated using Accutase and resuspended in staining buffer with 3% BSA and 0.5 mM EDTA. After fixation and permeabilization, the cells were incubated with the Click-iT reaction cocktail for 30 min at room temperature, protected from light. DAPI was added for DNA staining, and the cells were analyzed using a FACSAria II flow cytometer (BD Biosciences) and FCS Express 7 software (DeNovo Software).

## Bulk RNA sequencing and analysis

For FAM53C knock-down experiments in RPE-1 cells, $1 \times 10^6$ control and siRNA-treated samples were collected in duplicate 48 hr post-transfection, and RNA was isolated using the RNeasy Plus Micro kit

(QIAGEN #74034). Library preparation and sequencing were performed by Novogene using the Illumina NextSeq 500 platform to obtain ~20 million paired reads/sample. Raw sequencing reads were trimmed using CutAdapt (v2.10; *Martin, 2011*) with the TruSeq sequencing adapter (5'-AGATCGGA AGAGCACACGTCTGAACTCCAGTCAC-3'), and a minimum read length of at least 25. Reads were then aligned to the UCSC hg38 genome with HiSat2 (*Kim et al., 2019*) using reverse strandedness and discarding unaligned reads. Counts were assigned to genes (hg38 GTF) using featureCounts (*Liao et al., 2014*). Downstream differential expression analysis was performed using the Deseq2 package (*Love et al., 2014*). GO term enrichment of differentially expressed genes was performed using ClusterProfiler (*Wu et al., 2021*).

## FAM53C interactome analysis

### Tandem affinity purification

10 mL packed cell volume of HEK-293T cells expressing LAP-tagged proteins (*Torres et al., 2009*) were re-suspended with 40 mL of LAP-resuspension buffer (300 mM KCl, 50 mM HEPES-KOH [pH 7.4], 1 mM EGTA, 1 mM MgCl2, 10% glycerol, 0.5 mM DTT, protease inhibitor [A32965, Thermo Fisher Scientific]), lysed by gradually adding 1200 µL 10% NP-40 to a final concentration of 0.3%, then incubated on ice for 10 min. The lysate was first centrifuged at 14,000 rpm (27,000 × $g$) at 4 °C for 10 min, and the resulting supernatant was centrifuged at 43,000 rpm (100,000 × $g$) for 1 hr at 4 °C to further clarify the lysate. High-speed supernatant was mixed with 500 µL of GFP-coupled beads (*Torres et al., 2009*) and rotated for 1 hr at 4 °C to capture GFP-tagged proteins, then washed five times with 1 mL LAP200N buffer (200 mM KCl, 50 mM HEPES-KOH [pH 7.4], 1 mM EGTA, 1 mM MgCl2, 10% glycerol, protease inhibitors, and 0.05% NP40). After re-suspending the beads with 1 mL LAP200N buffer lacking protease inhibitors, the GFP tag was cleaved by adding 40 µg PreScission-protease and rotating tubes at 4 °C for 16 hr. PreScission-protease eluted supernatant was added to 100 µL of S-protein agarose (69704–3, EMD Millipore) to capture S-tagged protein. After washing three times with LAP200N buffer and twice with LAP0 buffer (50 mM HEPES-KOH [pH 7.4], 1 mM EGTA, 1 mM MgCl2, and 10% glycerol), purified protein complexes were eluted with 50 µL of 2 X LDS buffer and boiled at 95 °C for 3 min. Samples were then run on Bolt Bis-Tris Plus Gels (NW04120BOX, Thermo Fisher Scientific) in Bolt MES SDS Running Buffer (B0002, Thermo Fisher Scientific). Gels were fixed and stained according to Colloidal Blue Staining Kit (LC6025, Thermo Fisher Scientific) with Optima LC/MS grade water (W6-4, Thermo Fisher Scientific) at room temperature. The buffer was then replaced with Optima water prior to cutting the bands into eight pieces. The gel slices were then destained, reduced, and alkylated followed by in-gel digestion using (200 ng) Trypsin/LysC (V5073, Promega) as previously (*Shevchenko et al., 2006*). Tryptic peptides were extracted from the gel bands and dried in a speed vac. Prior to LC-MS, each sample was reconstituted in 0.1% formic acid, 2% acetonitrile, and water.

### Chemicals and reagents

The reagents used in this analysis were obtained from various sources. From Sigma-Aldrich, we obtained 2-Chloroacetamide (CAM; cat. no. C0267), and Potassium hydroxide (KOH; cat. no. P5958). Trifluoroacetic acid (TFA; cat. no. AAL06374AC), Acetic acid (AcOH; cat. no. A11350), Optima LC/MS Grade Water (cat. no. W6-4), and 99.5% Formic acid, LC/MS Grade (FA; cat. no. A117-50) were supplied by Thermo Fisher Scientific. Tris(2-carboxyethyl)phosphine hydrochloride (TCEP-HCl; cat. no. PG82080) and Halt Protease and Phosphatase Inhibitor Cocktails, EDTA-Free (cat. nos. 78425 and 78428, respectively), were purchased from Thermo Fisher Scientific. Honeywell provided the LC/MS grade acetonitrile (ACN; cat. no. 14261–1 L). Additionally, Trypsin/Lys-C Mix, Mass Spec Grade (cat. no. V5073) from Promega, and Empore C18 47 mm Extraction Disk (cat. no. 320907D) from Empore were used. Liquid chromatography was conducted using a Bruker PepSep C18 10 cm packed column with 1.5 µm beads and 150 µm I.D (cat. no. 1893483) attached to a ZDV Sprayer with 20 µm I.D (cat. no. 1865710).

### Stage-tips clean up

Homemade Stage-Tips were constructed using two C18 Empore disks, following established procedures. The fabricated Stage-Tips were subjected to a washing process, which involved two washes with 100 µL of methanol, one wash with 100 µL of 80% acetonitrile/0.1% acetic acid, and two washes

with 100 µL of 1% acetic acid. Acidified peptides were loaded onto the Stage-Tips. Subsequently, the Stage-Tips were washed three times with 100 µL of 1% acetic acid to remove salts. Finally, the peptides were eluted from the Stage-Tips using two elution steps of 30 µL each, with 60% acetonitrile/0.1% acetic acid as the elution buffer.

## Liquid chromatography setup

A nanoELute ultra-high-pressure nano-flow chromatography system was utilized and directly coupled online with a hybrid trapped ion mobility spectrometry—quadrupole time-of-flight mass spectrometer (timsTOF Pro, Bruker) employing a nano-electrospray ion source (CaptiveSpray, Bruker Daltonics).

## Chromatographic conditions

The liquid chromatography was conducted at a constant temperature of 50 °C, employing a reversed-phase column (PepSep column, 10 cm×150 µm i.d., packed with 1.5 µm C18-coated porous silica beads, Bruker) connected to the 10 µm emitter (Bruker). The mobile phase consisted of two components: Mobile Phase A, comprising water with 0.1/2% formic acid/ACN (v/v), and Mobile Phase B, comprising ACN with 0.1% formic acid (v/v).

## Gradient elution

Peptide separation was achieved using a linear gradient from 2–33% Mobile Phase B within 60 min. This was followed by a washing step with 95% Mobile Phase B and subsequent re-equilibration. The chromatographic process maintained the flow rate at 400 nL/min.

## MS acquisition

Samples were analyzed using the timsTOF HT Mass Spectrometer in DDA-PASEF mode. The TIMS elution voltage was calibrated linearly to obtain reduced ion mobility coefficients (1 /K0) by using three selected ions from the Agilent ESI-L Tuning Mix (m/z 622, 922, 1222). The mass and ion mobility ranges were set from 100 to 1700 m/z and 0.7–1.3 1 /K0, respectively. Both ramp and acquisition times were set at 100ms. Precursor ions suitable for PASEF-MS/MS were chosen from TIMS-MS survey scans using the PASEF scheduling algorithm. A polygon filter was applied to the m/z and ion mobility plane to prioritize features likely representing peptide precursors over singly charged background ions. The quadrupole isolation width was set to 2 Th for m/z<700 and 3 Th for m/z>700, with collision energy linearly increased from 20 to 60 eV as ion mobility ranged from 0.6 to 1.6 (1 /K0).

## Silver staining

5 µL of samples containing LDS buffer and DTT prepared for tandem affinity purification and mass spectrometry described above were mixed with 1.25 µL of 500 mM iodoacetamide (0210035105, MP Biomedicals). Proteins were separated in a 4–12% Bis-Tris gel (NP0321BOX, Invitrogen), followed by fixation of the gel overnight in 50% methanol at room temperature. The gel was impregnated with solution C (0.8% (w/v) silver nitrate (S6506, SIGMA), 207.2 mM ammonium hydroxide (A6899, SIGMA), and 18.9 mM sodium hydroxide) for 15 min, followed by rinsing with water twice. The image was then developed in solution D (0.005% citric acid and 0.0185% formaldehyde in Milli-Q) until the intensity of the bands increased to optimal level. The reaction was then terminated by adding stop solution (45% methanol and 10% acetic acid).

## Data analysis

Raw mass spec result files were processed using Bionic (Protein Metrics, Inc) software (version: 4.5.2) with the following parameters: precursor mass tolerance: 20 ppm, fragment mass tolerance: 40 ppm; fragmentation type: QTOF/HCD; carbamidomethyl as Cys fixed modification, variable modifications of Met oxidation, Asn and Gln deamidation, pyro-Glu formation at N-term Glu and Gln, N-term acetylation, and Ser, Thr, and Tyr phosphorylation, with a maximum of three variable modifications; trypsin with a maximum of two missed cleavages and fully specific mode; identifications were filtered at 0.01 FDR at the protein level; a fasta library of all human refseq proteins (curated and predicted) was used that was downloaded on July 20, 2021.

Spectral counts were normalized to NSAF values (*Zybailov et al., 2006*) and significance of enrichment of bait-association was calculated as described previously (*Ding et al., 2016*).

## Proteomic analysis

BioGRID (*Oughtred et al., 2021*). Gene ontology (GO) term enrichment analysis and protein-protein interaction analysis were performed using Metascape with the default settings in the 'Express Analysis' function. We used Enrichr (*Chen et al., 2013b*) to analyze the top candidate interactors from the IP/MS experiments (66 proteins, p-val<0.05) on 07/25/2024.

## Recombinant protein expression and purification

Recombinant proteins were expressed in *Escherichia coli* BL21 (DE3) cells. The cells were transformed with plasmids encoding the target protein fused with either a His-tag (FAM53C) or a GST-tag (DYRK1A). Transformed BL21 cells were cultured in LB medium supplemented with 100 µg/mL ampicillin. Protein expression was induced at an $OD_{600}$ of 0.8 using 1 mM IPTG. The cultures were grown overnight at 20 °C.

The bacterial cells were harvested by centrifugation and resuspended in lysis buffer containing 50 mM Tris-HCl pH 8.0, 150 mM NaCl, and 1 mM TCEP. The cells were lysed by cell homogenizer and clarified by centrifugation. For His-tagged proteins, the lysate was applied to a nickel-nitrilotriacetic acid (Ni-NTA) column pre-equilibrated with equilibration buffer (50 mM Tris-HCl pH 8.0, 150 mM NaCl, 1 mM TCEP, 10 mM imidazole). After incubation for 30 min, resin with bound protein was washed with the same buffer but containing 30 mM imidazole, and subsequently bound proteins were eluted with 200 mM imidazole buffer. For GST-tagged proteins, the lysate was applied to a GST-affinity resin pre-equilibrated with equilibration buffer (50 mM Tris-HCl pH 8.0, 150 mM NaCl, 1 mM TCEP), and the protein was eluted using the same buffer but containing 20 mM reduced glutathione.

The His or GST tag was cleaved by incubating the eluted protein with 0.1 mg/mL GST-TEV protease overnight at 4 °C. Following cleavage, the protein mixture was passed through a GST-affinity column to remove GST-TEV and cleaved tag. The tag-free protein was further purified by ion exchange chromatography using a Q Sepharose column. The protein solution was loaded onto the Q column pre-equilibrated with 50 mM Tris-HCl pH 8.0 and 50 mM NaCl. Bound proteins were eluted using a gradient of 50 mM to 1 M NaCl. As a final purification step, the protein was subjected to size exclusion chromatography on a Superdex 75 column equilibrated with buffer containing 50 mM Tris-HCl pH 8.0, 500 mM NaCl, and 1 mM TCEP. Fractions containing the target protein were pooled and concentrated using centrifugal filtration. The purity of the final protein preparation was confirmed by SDS-PAGE, and protein was stored at –80 °C in 10% glycerol.

To obtain biotinylated DYRK1A, we used sortase labeling as described (*Theile et al., 2013*), utilizing the N-terminal glycine that is left following TEV cleavage. 50 µM DYRK1A was reacted with 500 µM of a synthetic biotin-LPETGG peptide and a final concentration of 20 µM purified His-tagged sortase in a final reaction volume of 500 µL. The reaction was incubated overnight at 4 °C. Following the reaction, the DYRK1A was purified again by passing over $Ni^{2+}$-NTA and Superdex 75 columns and was stored as above. The Cyclin D1 used in the recombinant protein kinase assay is a complex of CDK4-CycD1 purified as previously described (*Guiley et al., 2019*).

## Kinase assays

Protein samples were mixed in a buffer containing 25 mM Tris-HCl pH 8.0, 150 mM NaCl, 20 mM $MgCl_2$, and 1 mM DTT. A mixture of non-radioactive and [γ- $P^{32}$] ATP (~10 µCi) was added to the assay samples at a final concentration of 200 nM and allowed to react for 15 or 30 min as indicated. The samples were subjected to SDS-PAGE for protein separation. The SDS-PAGE was performed on a 4–20% gradient at 200 V for 40 min under denaturing conditions. Following electrophoresis, the gel was dried down completely to capture the signal from the radiolabeled ATP. The radioactive decay was detected by exposing the dried gel to a phosphor screen overnight. The phosphor screen was scanned using a GE Typhoon Trio Imager.

## SM13797 selectivity profile

Biochemical $IC_{50}$ values for CLK1-CLK4, DYRK1A, CDK1, and GSK3β were determined by acoustically transferring SM13797 to 384-well plates (Echo 550, LabCyte) and by performing kinase assays using

ThermoFisher LanthaScreen platform for CLK4 or Z'LYTE platform for the other kinases following the manufacturer's instructions. $IC_{50}$ values were calculated from 11-point dose response curves using Dotmatics Studies software. The full kinome screen (484 kinases) was performed using ThermoFisher SelectScreen profiling service with SM13797 at 1 μM. The $IC_{50}$ values were subsequently determined for each kinase demonstrating >90% inhibition.

Target engagement $IC_{50}$ values were determined using the Promega NanoBRET TE Intracellular Kinase Assay platform in transiently transfected HEK293T cells. $IC_{50}$ values were determined from 10-point dose response curves using Dotmatics Studies software.

### Biolayer interferometry

BLI experiments were performed using an eight-channel Octet-RED96e (Santorius). Experiments were performed in an assay buffer containing 25 mM Tris pH 8.0, 150 mM NaCl, 1 mM DTT, 2 mg/mL BSA, and 0.2% (v/v) Tween. Samples were formatted in a 96-well plate with each well containing 200 μL. For each experiment, we used eight streptavidin biosensor tips (Santorius) that were loaded with the 200 nM biotinylated DYRK1A and dipped into FAM53C analyte at varied concentrations. All experiments were accompanied by reference measurements using unloaded streptavidin tips dipped into the same analyte-containing wells. Experiments in which analyte concentration was varied also contained a zero-analyte reference. Data were processed and fit using Octet software version 7 (Sartorius). Before fitting, all datasets were double reference-subtracted, aligned on the $y$ axis through their respective baselines, aligned for interstep correction through their respective dissociation steps, and smoothened using Savitzky–Golay filtering. The first 20 s of association and 20 s of dissociation were used with a 1:1 binding model. The association and dissociation rate constants as well as standard errors were averaged across the set of sensorgrams. The equilibrium dissociation constant $K_D$ was obtained from a steady state analysis of all concentrations of analyte fit to the equation Response = $(R_{max} * [Analyte])/(K_D + [Analyte])$.

### Statistical analysis

Statistical significance was assessed using Prism GraphPad software, unless otherwise stated in the methods above. The tests used are noted in the figure legends and accompanying Supplementary Files (paired t-tests were used unless otherwise stated). Data are represented as mean ± standard deviation unless otherwise stated.

## Acknowledgements

We thank all the members of the Sage lab for their help and support throughout this study. Research reported in this publication was supported by the NIH (JS, SMR, JMS, PKJ P01CA254867). JS is the Elaine and John Chambers Professor in Pediatric Cancer.

## Additional information

### Competing interests

Carine Bossard: C.B. is an employee of Biosplice Therapeutics, Inc. The other authors declare that no competing interests exist.

### Funding

| Funder | Grant reference number | Author |
| --- | --- | --- |
| National Cancer Institute | P01CA254867 | Jan M Skotheim<br>Peter K Jackson<br>Julien Sage |

The funders had no role in study design, data collection and interpretation, or the decision to submit the work for publication.

### Author contributions

Taylar Hammond, Conceptualization, Formal analysis, Investigation, Visualization, Writing – original draft, Writing – review and editing; Jong Bin Choi, Janos Demeter, Roy Ng, Formal

analysis, Investigation, Visualization, Writing – original draft, Writing – review and editing; Miles W Membreño, Investigation, Visualization, Writing – original draft, Writing – review and editing; Debadrita Bhattacharya, Griffin G Hartmann, Caterina I Colon, Formal analysis, Visualization, Writing – original draft, Writing – review and editing; Thuyen N Nguyen, Investigation, Writing – original draft, Writing – review and editing; Carine Bossard, Resources, Investigation, Visualization, Methodology, Writing – original draft, Project administration, Writing – review and editing; Jan M Skotheim, Conceptualization, Funding acquisition, Visualization, Writing – original draft, Project administration, Writing – review and editing; Peter K Jackson, Anca M Pasca, Conceptualization, Supervision, Visualization, Writing – original draft, Project administration, Writing – review and editing; Seth M Rubin, Conceptualization, Formal analysis, Supervision, Investigation, Visualization, Writing – original draft, Writing – review and editing; Julien Sage, Conceptualization, Formal analysis, Supervision, Funding acquisition, Visualization, Writing – original draft, Project administration, Writing – review and editing

### Author ORCIDs
Jong Bin Choi ⓘ https://orcid.org/0000-0002-7017-1775
Roy Ng ⓘ https://orcid.org/0009-0004-4325-4753
Debadrita Bhattacharya ⓘ https://orcid.org/0000-0002-1923-2675
Carine Bossard ⓘ https://orcid.org/0009-0005-2508-7282
Jan M Skotheim ⓘ https://orcid.org/0000-0001-8420-6820
Peter K Jackson ⓘ https://orcid.org/0000-0002-1742-2539
Anca M Pasca ⓘ https://orcid.org/0000-0002-0445-9009
Seth M Rubin ⓘ https://orcid.org/0000-0002-1670-4147
Julien Sage ⓘ https://orcid.org/0000-0002-8928-9968

### Ethics
Mice were maintained at Stanford University's Research Animal Facility according to practices prescribed by the NIH and by the Institutional Animal Care and Use Committee (IACUC) at Stanford University. Additional accreditation of Stanford University Research Animal Facility was provided by the Association for Assessment and Accreditation of Laboratory Animal Care (AAALAC). The study protocol was approved by the Administrative Panel on Laboratory Animal Care (APLAC) at Stanford University (protocol 13565).

Reviewer #1 (Public review): https://doi.org/10.7554/eLife.109708.3.sa1
Reviewer #3 (Public review): https://doi.org/10.7554/eLife.109708.3.sa2
Author response https://doi.org/10.7554/eLife.109708.3.sa3

---

# Additional files

### Supplementary files
Supplementary file 1. 38 genes in the G1 network and co-dependency scores for each.
Supplementary file 2. Genes shared in co-dependency analysis of 38 G1 genes.
Supplementary file 3. Candidate FAM53C interactors identified by immunoprecipitation followed by mass spectrometry analysis.
Supplementary file 4. Term enrichment analysis for top FAM53C candidate interactors.
Supplementary file 5. RNA-seq analysis of FAM53C knockdown in human RPE-1 cells.
MDAR checklist

### Data availability
Sequencing data from RNA sequencing are available from the Gene Expression Omnibus (GEO) under accession numbers GSE282945. The mass spectrometry proteomics data have been deposited to the ProteomeXchange Consortium via the PRIDE (*Vizcaíno et al., 2016*) repository with the dataset identifier PXD055829 (10.6019/PXD055829). Data related to the IPMC genotyping and phenotyping of Fam53C mutant mice is available on the IPMC web sitewebsite. All other data and new material are available in the article and supplementary materials, at this link https://doi.org/10.5281/zenodo.19712495 (for source data), or from the corresponding author upon reasonable request.

The following datasets were generated:

| Author(s) | Year | Dataset title | Dataset URL | Database and Identifier |
|---|---|---|---|---|
| Hammond T, Bhattacharya D, Sage J | 2025 | The FAM53C/DYRK1A axis regulates the G1/S transition of the cell cycle | https://www.ncbi.nlm.nih.gov/geo/query/acc.cgi?acc=GSE282945 | NCBI Gene Expression Omnibus, GSE282945 |
| Demeter J, Jackson PK | 2026 | The FAM53C/DYRK1A Axis Regulates The G1/S Transition Of The Cell Cycle | https://www.ebi.ac.uk/pride/archive/projects/PXD055829 | PRIDE, PXD055829 |
| Sage J | 2026 | Source Data for publication Hammond et al. | https://doi.org/10.5281/zenodo.19712496 | Zenodo, 10.5281/zenodo.19712496 |

The following previously published dataset was used:

| Author(s) | Year | Dataset title | Dataset URL | Database and Identifier |
|---|---|---|---|---|
| Arafeh R, Shibue T, Dempster JM | 2025 | DepMap | https://doi.org/10.1038/s41568-024-00763-x | Depmap.org, 10.1038/s41568-024-00763-x |

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
