## [Editor Report · eLife Assessment]

This study identifies the uncharacterised protein FAM53C as a novel, potential regulator of the G1/S cell cycle transition, linking its function to the DYRK1A kinase and the RB/p53 pathways. The work is **valuable** and of interest to the cell cycle field, leveraging a strong computational screen to identify a new candidate. The findings are **solid**, although confidence in the siRNA depletion phenotypes would have been higher with rescue experiments using an siRNA-resistant cDNA.

[Editors' note: this paper was reviewed by Review Commons.]

---

## [Referee Report · Reviewer #1 (Public review)]

[Editors' note: This version has been assessed by the Reviewing Editor without further input from the original reviewers. The authors have addressed comments raised in the previous round of review, shown below, through minor changes to the text without additional experiments.]

Summary:

Taylar Hammond and colleagues identified new regulators of the G1/S transition of the cell cycle. They did so by screening publicly available data from the Cancer Dependency Map and identified FAM53C as a positive regulator of the G1/S transition. Using biochemical assays they then show that FAM53 interacts with the DYRK1A kinase to inhibit its function. They show in RPE1 cells that loss of FAMC53 leads to a DYRK1A + P53-dependent cell cycle arrest. Combined inactivation of FAM53C and DYRK1A in a TP53-null background caused S-phase entry with subsequent apoptosis. Finally the authors assess the effect of FAM53C deletion in a cortical organoid model, and in Fam53c knockout mice. Whereas proliferation of the organoids is indeed inhibited, mice show virtually no phenotype.

Reviewer #2 (Public review):

The authors sought to identify new regulators of the G1/S transition by mining the Cancer Dependency Map (DepMap) co-dependency dataset. This analysis successfully identified FAM53C, a poorly characterized protein, as a candidate. The strength of the paper lies in this initial discovery and the subsequent biochemical work convincingly showing that FAM53C can directly interact with the kinase DYRK1A, a known cell cycle regulator.

The authors then present evidence, primarily from acute siRNA knockdown in RPE-1 cells, that loss of FAM53C induces a strong G1 cell cycle arrest. Their follow-up investigation proposes a model where FAM53C normally inhibits DYRK1A, thereby protecting Cyclin D from degradation and preventing p53 activation, to allow for G1/S progression. The authors have commendably addressed some concerns from the initial review: they have now demonstrated the G1 arrest using two independent siRNAs (an improvement over the initial pool), shown the effect in several additional cancer cell lines (U2OS, A549, HCT-116), and developed a more nuanced model that incorporates p53 activation, which helps to explain some of the complex data.

---

## [Referee Report · Reviewer #3 (Public review)]

In this study Hammond et al. investigated the role of Dual-specificity Tyrosine Phosphorylation regulated Kinase 1A (DYRK1) in G1/S transition. By exploiting Dependency Map portal, they identified a previously unexplored protein FAM53C as potential regulator of G1/S transition. Using RNAi, they confirmed that depletion of FAM53C suppressed proliferation of human RPE1 cells and that this phenotype was dependent on the presence protein RB. In addition, they noted increased level of CDKN1A transcript and p21 protein that could explain G1 arrest of FAM53C-depleted cells but surprisingly, they did not observe activation of other p53 target genes. Proteomic analysis identified DYRK1 as one of the main interactors of FAM53C and the interaction was confirmed in vitro. Further, they showed that purified FAM53C blocked the ability of DYRK1 to phosphorylate cyclin D in vitro although the activity of DYRK1 was likely not inhibited (judging from the modification of FAM53C itself). Instead, it seems more likely that FAM53C competes with cyclin D in this assay. Authors claim that the G1 arrest caused by depletion of FAM53C was rescued by inhibition of DYRK1 but this was true only in cells lacking functional p53. This is quite confusing as DYRK1 inhibition reduced the fraction of G1 cells in p53 wild type cells as well as in p53 knock-outs, suggesting that FAM53C may not be required for regulation of DYRK1 function. Instead of focusing on the impact of FAM53C on cell cycle progression, authors moved towards investigating its potential (and perhaps more complex) roles in differentiation of IPSCs into cortical organoids and in mice. They observed a lower level of proliferating cells in the organoids but if that reflects an increased activity of DYRK1 or if it is just an off-target effect of the genetic manipulation remains unclear. Even less clear is the phenotype in FAM53C knock-out mice. Authors did not observe any significant changes in survival nor in organ development but they noted some behavioral differences. Whether and how these are connected to the rate of cellular proliferation was not explored. In the summary, the study identified previously unknown role of FAM53C in proliferation but failed to explain the mechanism and its physiological relevance at the level of tissues and organism.

Comments on the previous version:

In the revised version of the manuscript, authors addressed most of the critical points. They now include new data with depletion of FAM53C using single siRNAs that show small but significant enrichment of population of the G1 cells. This G1 arrest is likely caused by a combined effects on induction of p21 expression and decreased levels of cyclin D1. Authors observed that inhibition of DYRK1 rescued cyclin D1 levels in FAM53 depleted cells suggesting that FAM53C may inhibit DYRK1. This possibility is also supported by in vitro experiments. On the other hand, inhibition of DYRK1 did not rescue the G1 arrest upon depletion of FAM53C, suggesting that FAM53C may have also DYRK1-independent role in G1. Functional rescue experiments with cyclin D1 mutants and detection of DYRK1 activity in cells would be necessary to conclusively explain the function of FAM53C in progression through G1 phase but unfortunately these experiments were technically not possible. Knock out of FAM53C in iPSCs and in mice suggest that FAM53C may have additional functions besides the cell cycle control and/or that adaptation may have occurred in these model systems. Overall, the study implicated FAM53C in fine tuning DYRK1 activity in cells that may to some extent influence the progression through G1 phase. In addition, FAM53C may also have DYRK1 and cell cycle independent functions that remain to be addressed by future studies.

---

## [Author Response]

The following is the authors’ response to the original reviews

**Public Reviews:**

**Reviewer #1 (Public review):**
Summary:Taylar Hammond and colleagues identified new regulators of the G1/S transition of the cell cycle. They did so by screening publicly available data from the Cancer Dependency Map and identified FAM53C as a positive regulator of the G1/S transition. Using biochemical assays they then show that FAM53 interacts with the DYRK1A kinase to inhibit its function. They show in RPE1 cells that loss of FAMC53 leads to a DYRK1A + P53-dependent cell cycle arrest. Combined inactivation of FAM53C and DYRK1A in a TP53-null background caused S-phase entry with subsequent apoptosis. Finally the authors assess the effect of FAM53C deletion in a cortical organoid model, and in Fam53c knockout mice. Whereas proliferation of the organoids is indeed inhibited, mice show virtually no phenotype.The authors have revised the manuscript, and I respond here point-by-point to indicate which parts of the revision I found compelling, and which parts were less convincing. So the numbering is consistent with the numbering in my first review report.(1) The p21 knockdowns are a valuable addition, and the claim that other p53 targets than p21 are involved in the FAMC53 RNAi-mediated arrest is now much more solid. Minor detail: if S4D is a quantification of S4C, it is hard to believe that the quantification was done properly (at least the DYRK1Ai conditions). Perhaps S4C is not the best representative example, or some error was made?

We appreciate the concern from the Reviewer. As explained in the first round of revisions, we have mostly used an immunoassay based on capillary transfer (WES system), which is very quantitative (much more than classical immunoblot). As for the other WES assays, the panel in S4C is a representation from the signal in the capillary from one of the experiments we performed (in many ways, we should simply not show these representations but readers and reviewers expect them). We agree that this was not visually the most representative, likely because of the saturation of the signal, and we replaced it with another one.

(2a) I appreciate the decision to remove the cyclin D1 phosphorylation data. A more nuanced model now emerges. It is not clear to me however why the Protein Simple immunoassay was used for experiments with RPE cells, and not the cortical organoids. Even though no direct claims are made based on the phospho-cyclin D data in Figure 5E+G, showing these data suggests that FAM53C deletion increases DYRK1A-mediated cyclin D1 phosphorylation. I find it tricky to show these data, while knowing now that this effect could not be shown in the RPE1 cells.

The Reviewer raises a valid point. The data we had presented in the first version of the manuscript were strongly suggestive of changes in Cyclin D1 phosphorylation and protein stability but we followed the Reviewer’s advice to remove them from the revised manuscript because the effects were sometimes small. We decided to keep these data in the organoid model because we felt this is a question that many readers would have (how do changes in FAM53C affect Cyclin D levels?). As the Reviewer mentions, we did not draw conclusions about this but we felt and still feel it is important to connect the dots, even if imperfectly, between FAM53C and the cell cycle, and these data in Figure complement the data in Figure 3F. The experiments with RPE-1 cells were mostly performed in the Sage lab with the WES assay while the experiments with organoids were largely performed in the Pasca lab where more ‘classic’ immunoblots are routinely used. More generally, some antibodies work better with one method vs. the other and we often go back and forth between the two.

(2b) The quantifications of the immunoassays are not convincing. In multiple experiments, the HSP90 levels vary wildly, which indicates big differences in protein loading if HSP90 is a proper loading control. This is for example problematic for the interpretation of figure 3F and S3I. The cyclin D1 "bands" look extremely similar between siCtrl and siFAM53C (Fig S3I), in fact the two series of 6 samples with different dosages of DYRK1Ai look seem an identical repetition of each other. I did not have to option to overlay them, but it would be important to check if a mistake was made here. The cyclin D1 signals aside, the change in cycD1/HSP90 ratios seems to be entirely caused by differences in HSP90 levels. Careful re-analysis of the raw data and more equal loading seem necessary. The same goes (to a lesser extent) for S3J+K.

As mentioned above, the representation of the fluorescence signal may be important for readers who are used to seeing immunoblot (Western blots), but the quantification is performed on the values directly obtained from the WES system from ProteinSimple. In these experiments, we make sure that the numbers we obtain are in a validated range, allowing us to use the values, even if sometimes the loading is a bit different between lanes. The sensitivity of the WES assay allows for high accuracy in intra-well quantification allowing for accurate inter-well quantification once loading control normalization is completed.

(2c) the new model in Fig S4L: what do the arrows at the right FAM53C and p53 that merge a point straight towards S-phase mean? They suggest that p53 (and FAM53C) directly promote S-phase progression, but most likely this is not what the authors intended with it.

Very good point. We were trying to be inclusive of various signaling pathways that may be implicated in the regulation of the cell cycle by this group of proteins. FAM53C does promote S-phase entry (more cycling when FAM53C is overexpressed) but we removed the arrow coming from p53, which is certainly not a positive regulator of cell cycle progression. Thank you for helping us correct this mistake.

(3) Clear; nicely addressed.(4) Thank you for correcting.(5) I appreciate that the authors are now more careful to call the IMPC analysis data preliminary. This is acceptable to me, but nevertheless, I suggest the authors to seriously consider taking this part entirely out. The risk of chance finding and the extremely skewed group sizes (as reviewer #2 had pointed out) hamper the credibility of this statistical analysis.

We appreciate this concern but feel that it is important for the community to be aware of these phenotypes so other investigators either study FAM53C in different genetic contexts or, for example, generate a conditional knockout allele to study more acute effects of FAM53C loss during development and in adult mice. We believe that the text is carefully written and acknowledge the caveats of small sample sizes in some statistical analyses.

**Reviewer #2 (Public review):**
The authors sought to identify new regulators of the G1/S transition by mining the Cancer Dependency Map (DepMap) co-dependency dataset. This analysis successfully identified FAM53C, a poorly characterized protein, as a candidate. The strength of the paper lies in this initial discovery and the subsequent biochemical work convincingly showing that FAM53C can directly interact with the kinase DYRK1A, a known cell cycle regulator.The authors then present evidence, primarily from acute siRNA knockdown in RPE-1 cells, that loss of FAM53C induces a strong G1 cell cycle arrest. Their follow-up investigation proposes a model where FAM53C normally inhibits DYRK1A, thereby protecting Cyclin D from degradation and preventing p53 activation, to allow for G1/S progression. The authors have commendably addressed some concerns from the initial review: they have now demonstrated the G1 arrest using two independent siRNAs (an improvement over the initial pool), shown the effect in several additional cancer cell lines (U2OS, A549, HCT-116), and developed a more nuanced model that incorporates p53 activation, which helps to explain some of the complex data.However, a central and critical weakness persists. The entire functional model is built upon the very strong G1 arrest phenotype observed in vitro following acute knockdown. This finding is in stark contrast to data from other contexts. As the authors note, the knockout of Fam53c in mice results in minimal phenotypes, and the DepMap data itself suggests the gene is largely non-essential in most cancer cell lines.This major discrepancy creates two competing interpretations:As the authors suggest, FAM53C has a critical role in the cell cycle, but its loss is rapidly masked by compensatory mechanisms in long-term knockout models (like iPSCs and mice) or in established cancer cell lines.The strong acute G1 arrest is an experimental artifact of the siRNA-mediated knockdown, and not a true reflection of FAM53C's primary function.The authors' new controls (using two individual siRNAs and showing the arrest is RB-dependent) make an off-target effect less likely, but they do not definitively rule it out. The gold-standard experiment to distinguish between these two possibilities-a rescue of the phenotype using an siRNA-resistant cDNA-has not been performed.Because this key control is missing, the foundation of the paper's functional claims is not as solid as it needs to be. While the study provides an interesting and valuable new candidate for the cell cycle field to investigate, readers should be cautious in accepting the strength of FAM53C's role in the G1/S transition until this central discrepancy is definitively resolved.

We appreciate this concern from the Reviewer. Genetically, *FAM53C* is linked to a number of genes coding for known regulators of the G1/S transition and its loss of function would be predicted to lead to G1 arrest based on these genetic interactions. As the Reviewer nicely summarizes, we have data in several cell types, including non-cancerous immortalized cells (RPE-1) and several cancer cell lines, that FAM53C acute knock-down leads to a G1 arrest. Our data also indicate that this arrest is RB dependent and p53 independent. Furthermore, genetic knockout of FAM53C in iPSC-derived human cortical organoids results in decreased proliferation. All these elements point to a role for FAM53C in G1/S. We performed some pilot rescue experiments, as suggested by the Reviewer, but these preliminary assays could not identify the right “dose” of FAM53C. We agree that it will be important in future studies to develop better genetic systems in which FAM53C can be manipulated genetically. However, our overexpression experiments show increased proliferation, providing more support for a role of FAM53C at the G1/S transition of the cell cycle.

**Reviewer #3 (Public review):**
Summary:In this study Hammond et al. investigated the role of Dual-specificity Tyrosine Phosphorylation regulated Kinase 1A (DYRK1) in G1/S transition. By exploiting Dependency Map portal, they identified a previously unexplored protein FAM53C as potential regulator of G1/S transition. Using RNAi, they confirmed that depletion of FAM53C suppressed proliferation of human RPE1 cells and that this phenotype was dependent on the presence protein RB. In addition, they noted increased level of CDKN1A transcript and p21 protein that could explain G1 arrest of FAM53C-depleted cells but surprisingly, they did not observe activation of other p53 target genes. Proteomic analysis identified DYRK1 as one of the main interactors of FAM53C and the interaction was confirmed in vitro. Further, they showed that purified FAM53C blocked the ability of DYRK1 to phosphorylate cyclin D in vitro although the activity of DYRK1 was likely not inhibited (judging from the modification of FAM53C itself). Instead, it seems more likely that FAM53C competes with cyclin D in this assay. Authors claim that the G1 arrest caused by depletion of FAM53C was rescued by inhibition of DYRK1 but this was true only in cells lacking functional p53. This is quite confusing as DYRK1 inhibition reduced the fraction of G1 cells in p53 wild type cells as well as in p53 knock-outs, suggesting that FAM53C may not be required for regulation of DYRK1 function. Instead of focusing on the impact of FAM53C on cell cycle progression, authors moved towards investigating its potential (and perhaps more complex) roles in differentiation of IPSCs into cortical organoids and in mice. They observed a lower level of proliferating cells in the organoids but if that reflects an increased activity of DYRK1 or if it is just an off-target effect of the genetic manipulation remains unclear. Even less clear is the phenotype in FAM53C knock-out mice. Authors did not observe any significant changes in survival nor in organ development but they noted some behavioral differences. Weather and how these are connected to the rate of cellular proliferation was not explored. In the summary, the study identified previously unknown role of FAM53C in proliferation but failed to explain the mechanism and its physiological relevance at the level of tissues and organism. Although some of the data might be of interest, in current form the data is too preliminary to justify publication.Major comments:(1) Whole study is based on one siRNA to Fam53C and its specificity was not validated. Level of the knock down was shown only in the first figure and not in the other experiments. The observed phenotypes in the cell cycle progression may be affected by variable knock-down efficiency and/or potential off target effects.

We fully acknowledge these limitations in our study. First, we agree that the efficiency of the knock-down can be variable across experiments; unfortunately, antibodies against FAM53C are currently still not optimal and immunoassays against this protein have not always been reliable in our hands. It will be important in the future to develop better antibodies for this poorly studied factor. Second, we also agree that the siRNA pool is perhaps not optimal (note that we used a pool, not a single siRNA). We provide data in the manuscript that single siRNAs (from the pool) also arrest cells in G1. Our data also show that this arrest in observed in several cell lines (cancerous and not cancerous), in a p53 independent but RB dependent way. We further note that we also provide data in cortical spheroids derived from CRISPR/Cas9 knockout iPSCs showing a similar inhibition of proliferation, validating our observations in a completely orthogonal system. Finally, overexpression studies support a role for FAM53C at the G1/S transition (i.e., FAM53C overexpression is sufficient to promote proliferation).

(2) Experiments focusing on the cell cycle progression were done in a single cell line RPE1 that showed a strong sensitivity to FAM53C depletion. In contrast, phenotypes in IPSCs and in mice were only mild suggesting that there might be large differences across various cell types in the expression and function of FAM53C. Therefore, it is important to reproduce the observations in other cell types.

As mentioned above, we have observed cell cycle arrest in several cancer cell lines (U2OS, A549, HCT-116) and in iPSC-derived organoids. We acknowledge that RPE-1 cells seem most sensitive to the knock-down and, currently, we do not understand why. In the future, it will be critical to gain a better understanding of the cellular/genetic contexts in which FAM53C plays more important roles in the G1/S transition; it will be also critical to understand what mechanisms may compensate for loss of FAM53C in cells, in culture and in vivo.

(3) Authors state that FAM53C is a direct inhibitor of DYRK1A kinase activity (Line 203), however this model is not supported by the data in Fig 4A. FAM53C seems to be a good substrate of DYRK1 even at high concentrations when phosphorylations of cyclin D is reduced. It rather suggests that DYRK1 is not inhibited by FAM53C but perhaps FAM53C competes with cyclin D. Further, authors should address if the phosphorylation of cyclin D is responsible for the observed cell cycle phenotype. Is this Cyclin D-Thr286 phosphorylation, or are there other sites involved?

We completely agree with the Reviewer that the functional interactions between FAM53C and DYRK1A will need to be explored further. Our data (and other data from mass spectrometry experiments in other contexts) support a model in which FAM53C binds to DYRK1A. Genetics analyses indicate that FAM53C is antagonistic to DYRK1A function. Our phosphorylation assays show decreased DYRK1A activity when FAM53C is present. Because our data also show that DYRK1A phosphorylates FAM53C, there may be more than one level of functional interaction between the two proteins, including effects by DYRK1A on FAM53C through its phosphorylation activity. We state in the text that our data suggest “that FAM53C may be a competitive substrate and/or an inhibitor of DYRK1A”, and we agree that we cannot provide a stronger conclusion at this point.

We believe that genetic data from DepMap and our data support a model in which Cyclin D is downstream of FAM53C in its regulation of the G1/S progression. As discussed with Reviewer #1, it has proven challenging to investigate how FAM53C may control the phosphorylation and degradation of Cyclin D. Thr286 is certainly a critical phosphorylation site, and this residue can be phosphorylated by DYRK1A, but whether FAM53C and DYRK1A engage with other residues or domains is not known and should be the focus of future studies.

(4) At many places, information on statistical tests is missing and SDs are not shown in the plots. For instance, what statistics was used in Fig 4C? Impact of FAM53C on cyclin D phosphorylation does not seem to be significant. In the same experiment, does DYRK1 inhibitor prevent modification of cyclin D?

We thank the Reviewer for this comment. We made sure in the revised version to mention all the statistical tests used.

(5) Validation of SM13797 compound in terms of specificity to DYRK1 was not performed.

We provided tables in Figure S3 that summarize the biochemical characterization of this DYRK1A inhibitor (performed by Biosplice Therapeutics, where this compound was developed)

(6) A fraction of cells in G1 is a very easy readout but it does not measure progression through the G1 phase. Extension of the S phase or G2 delay would indirectly also result in reduction of the G1 fraction. Instead, authors could measure the dynamics of entry to S phase in cells released from a G1 block or from mitotic shake off.

This is an interesting point raised by the Reviewer. It is correct that we only performed a more in-depth characterization of cell cycle phenotypes in certain contexts (e.g., cell counting, EdU incorporation) (see Figures 1 and S1). It is possible that different cell types adapt differently to loss or overexpression of FAM53C, and assays to synchronize the cells, including by mitotic shake off, maybe useful in future experiments to further characterize the cell cycle of FAM53C mutant cells.

Comments to the revised manuscript:In the revised version of the manuscript, authors addressed most of the critical points. They now include new data with depletion of FAM53C using single siRNAs that show small but significant enrichment of population of the G1 cells. This G1 arrest is likely caused by a combined effects on induction of p21 expression and decreased levels of cyclin D1. Authors observed that inhibition of DYRK1 rescued cyclin D1 levels in FAM53 depleted cells suggesting that FAM53C may inhibit DYRK1. This possibility is also supported by in vitro experiments. On the other hand, inhibition of DYRK1 did not rescue the G1 arrest upon depletion of FAM53C, suggesting that FAM53C may have also DYRK1-independent role in G1. Functional rescue experiments with cyclin D1 mutants and detection of DYRK1 activity in cells would be necessary to conclusively explain the function of FAM53C in progression through G1 phase but unfortunately these experiments were technically not possible. Knock out of FAM53C in iPSCs and in mice suggest that FAM53C may have additional functions besides the cell cycle control and/or that adaptation may have occurred in these model systems. Overall, the study implicated FAM53C in fine tuning DYRK1 activity in cells that may to some extent influence the progression through G1 phase. In addition, FAM53C may also have DYRK1 and cell cycle independent functions that remain to be addressed by future studies.
**Recommendations for the authors:**

**Reviewer #1 (Recommendations for the authors):**
All my minor points (6-11) were addressed adequately. No further comments.
**Reviewer #2 (Recommendations for the authors):**
The paper's conclusions would be substantially strengthened and the primary concern about off-target effects could be definitively resolved by performing one of the following two experiments:(1) Perform a rescue experiment. This would involve transfecting RPE-1 cells with an expression vector for an siRNA-resistant FAM53C cDNA (alongside a control vector) and then treating the cells with the FAM53C siRNAs. If the G1 arrest is a true on-target effect, the cells expressing the resistant cDNA should be "rescued" and continue to proliferate, while the control cells arrest. This is the most direct and standard way to validate a phenotype derived from siRNA.(2) Use an acute gene deletion approach that bypasses siRNAs entirely. The authors could use a lentiviral gRNA/Cas9 system to induce acute knockout of FAM53C in RPE-1 cells and assess the cell cycle phenotype at an early time point (e.g., 48-72 hours post-infection). This would provide a direct comparison to the acute siRNA knockdown, and if it recapitulates the strong G1 arrest, it would confirm the phenotype is due to FAM53C loss and not an artifact of the RNAi machinery. The current knockout models (iPSC, mice) are stable and long-term, which allows for the compensatory mechanism argument; an acute knockout would be a much stronger control. The authors could then also follow the fate of the cells and determine the nature of the suspected compensatory mechanisms.Addressing this central point is critical for the credibility of the proposed G1/S control element.

As discussed above, the observations of similar phenotypes in four cell lines (RPE-1 cells and three cancer cell lines) using a pool of siRNAs and in cortical organoids derived from iPSCs using a knockout approach strongly support our results. But we agree that our current study has limitations, including the lack of genetic re-introduction of FAM53C in knock-down or mutant cells. We also note that strong genetic evidence points to a role for FAM53C at the G1/S transition. We hope that some of the readers will be excited by FAM53C as an understudied factor with possible critical roles in fundamental cell biology and human diseases, and future studies will continue to investigate its function in cells using additional approaches.